# Identification of RBPMS as a mammalian smooth muscle master splicing regulator via proximity of its gene with super-enhancers

Erick E Nakagaki-Silva[1], Clare Gooding[1], Miriam Llorian[1,2], Aishwarya G Jacob[1,3], Frederick Richards[1], Adrian Buckroyd[1], Sanjay Sinha[1,3], Christopher WJ Smith[1]*

[1]Department of Biochemistry, University of Cambridge, Cambridge, United Kingdom; [2]Francis Crick Institute, London, United Kingdom; [3]Anne McLaren Laboratory, Cambridge Stem Cell Institute, University of Cambridge, Cambridge, United Kingdom

**Abstract** Alternative splicing (AS) programs are primarily controlled by regulatory RNA-binding proteins (RBPs). It has been proposed that a small number of master splicing regulators might control cell-specific splicing networks and that these RBPs could be identified by proximity of their genes to transcriptional super-enhancers. Using this approach we identified RBPMS as a critical splicing regulator in differentiated vascular smooth muscle cells (SMCs). RBPMS is highly down-regulated during phenotypic switching of SMCs from a contractile to a motile and proliferative phenotype and is responsible for 20% of the AS changes during this transition. RBPMS directly regulates AS of numerous components of the actin cytoskeleton and focal adhesion machineries whose activity is critical for SMC function in both phenotypes. RBPMS also regulates splicing of other splicing, post-transcriptional and transcription regulators including the key SMC transcription factor Myocardin, thereby matching many of the criteria of a master regulator of AS in SMCs.

*For correspondence:
cwjs1@cam.ac.uk

Competing interests: The authors declare that no competing interests exist.

## Introduction

Alternative splicing (AS) is an important component of regulated gene expression programmes during cell development and differentiation, usually focusing on different sets of genes than transcriptional control (*Blencowe, 2006*). AS programs re-wire protein-protein interaction networks (*Buljan et al., 2012*; *Ellis et al., 2012*; *Yang et al., 2016*), as well as allowing quantitative regulation by generating mRNA isoforms that are differentially regulated by translation or mRNA decay (*McGlincy and Smith, 2008*; *Mockenhaupt and Makeyev, 2015*). Coordinated cell-specific splicing programs are determined by a combination of *cis*-acting transcript features and *trans*-acting factors that compose 'splicing codes' (*Barash et al., 2010*; *Chen and Manley, 2009*; *Fu and Ares, 2014*). The interactions between the *trans* component RNA-binding proteins (RBPs) and the *cis* component regulatory elements in target RNAs coordinate the activation and repression of specific splicing events. Many regulatory proteins, including members of the SR and hnRNP protein families, are quite widely expressed, while others are expressed in a narrower range of cell types (*David and Manley, 2008*; *Fu and Ares, 2014*). A further conceptual development of combinatorial models for splicing regulation has been the suggestion that a subset of RBPs act as master regulators of cell-type specific AS networks (*Jangi and Sharp, 2014*). The criteria expected of such master regulators include that: (i) they are essential for cell-type specification or maintenance, (ii) their direct and indirect targets are important for cell-type function, (iii) they are likely to regulate the activity of other splicing regulators, (iv) they exhibit a wide dynamic range of activity, which is not limited by

**eLife digest** All the cells in our body contain the same genetic information, but they only switch on the genes that they need to fulfill their specific role in the organism. Genetic sequences known as enhancers can turn on the genes that are required by a particular cell to perform its tasks.

Once a gene is activated, its sequence is faithfully copied into a molecule of RNA which contains segments that code for a protein. A molecular machine then processes the RNA molecule and splices together the coding segments. RNA binding proteins can also regulate this mechanism, and help to splice the coding sections in different ways depending on the type of cell. The process, known as alternative RNA splicing, therefore creates different RNA templates from the same gene. This gives rise to related but different proteins, each suited to the needs of the particular cell in which they are made.

However, in some cell types, exactly how this happens has not yet been well documented. For example, in cells that line blood vessels – known as vascular smooth muscle cells – the RNA binding proteins that drive alternative splicing have not been identified.

To find these proteins, Nakagaki-Silva et al. used catalogs of DNA regions called super-enhancers as clues. These sequences strongly activate certain genes in a tissue-specific manner, effectively acting as labels for genes important for a given cell type. In vascular smooth muscle cells, if a super-enhancer switches on a gene that codes for a RNA-binding protein, this protein is probably crucial for the cell to work properly.

The approach highlighted a protein called RBPMS, and showed that it controlled alternative RNA splicing of many genes important in smooth muscle cells. This may suggest that when RBPMS regulation is disrupted, certain diseases of the heart and blood vessels could emerge. Finally, the results by Nakagaki-Silva et al. demonstrate that super-enhancers can signpost genes important in regulating splicing or other key processes in particular cell types.

autoregulation, and (v) they are regulated externally from the splicing network, for example by transcriptional control or post-translational modifications. It was further suggested that expression of such splicing master regulators would be driven by transcriptional super-enhancers, providing a possible route to their identification (*Jangi and Sharp, 2014*). Super-enhancers are extended clusters of enhancers that are more cell-type-specific than classical enhancers and that drive expression of genes that are essential for cell-type identity, including key transcription factors (*Hnisz et al., 2013*). By extension, RBPs whose expression is driven by super-enhancers are expected to be critical for cell-type identity and might include master regulators of tissue-specific AS networks (*Jangi and Sharp, 2014*).

Vascular smooth muscle cells (SMCs) are important in cardiovascular physiology and pathology (*Bennett et al., 2016*; *Fisher, 2010*; *Owens et al., 2004*). Unlike skeletal and cardiac muscle SMCs exhibit phenotypic plasticity and are not terminally differentiated (*Owens et al., 2004*) (*Figure 1A*). In healthy arteries, vascular SMCs exist in a differentiated contractile state. In response to injury or disease, the SMC phenotype switches towards a more synthetically active, motile and proliferative state (*Fisher, 2010*; *Owens et al., 2004*). The transcriptional control of SMC phenotypic switching has been intensely studied, but the role of post-transcriptional regulation has been relatively neglected (*Fisher, 2010*). For example, some markers of the contractile state, such as h-Caldesmon and meta-Vinculin, arise via AS (*Owens et al., 2004*), but nothing is known about the regulation of these events. A number of known splicing regulators, including PTBP1, CELF, MBNL, QKI, TRA2B, and SRSF1, have been implicated in the regulation of individual SMC-specific ASEs, but these proteins are not restricted to differentiated SMCs and most act primarily in the de-differentiated state (*Gooding et al., 2013*; *Gooding et al., 1998*; *Gromak et al., 2003*; *Shukla and Fisher, 2008*; *van der Veer et al., 2013*; *Xie et al., 2017*). Indeed, global profiling confirmed a widespread role of PTBP1 in repressing exons that are used in differentiated mouse aorta SMCs (*Llorian et al., 2016*), but did not identify RBPs that act as direct regulators of the differentiated state. Biochemical identification of such RBPs is hampered by the fact that SMCs rapidly dedifferentiate in cell culture conditions.

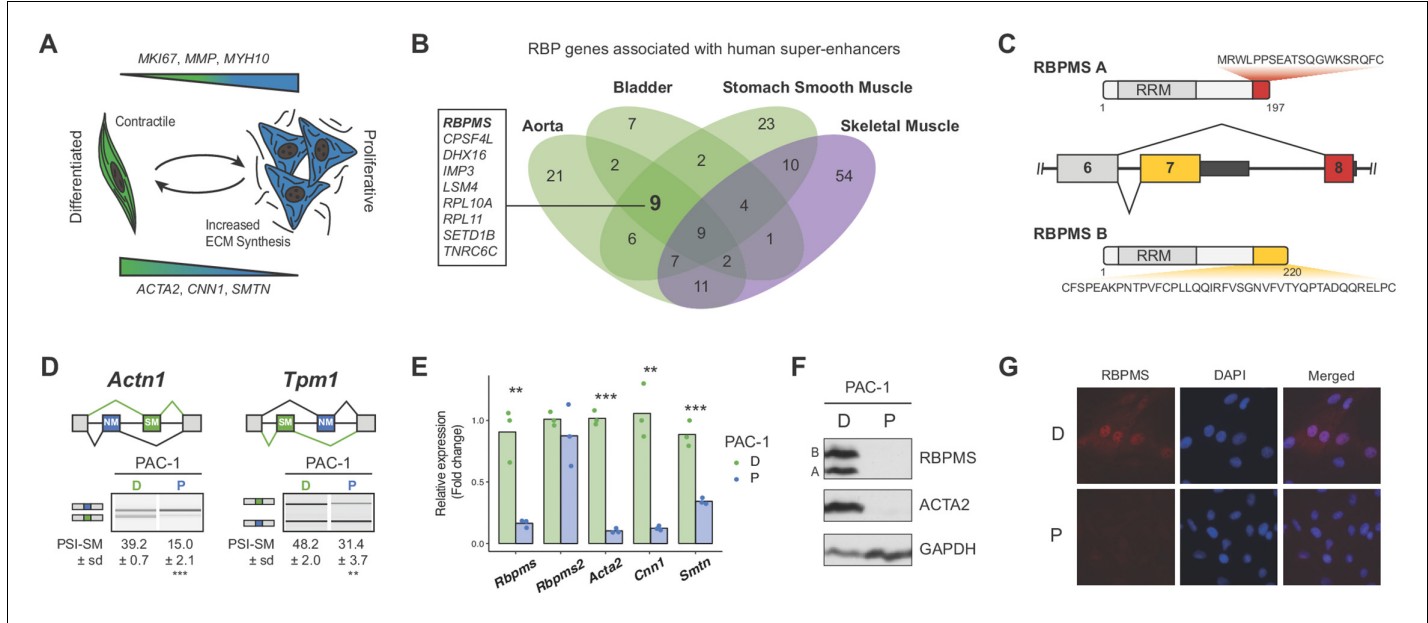

**Figure 1.** RBPMS is associated with SMC super-enhancers and is highly expressed in the differentiated PAC1 cells. (**A**) Diagram of the SMC dedifferentiation. SMC markers of differentiation and dedifferentiation are respectively shown at the bottom and top of the diagram. (**B**) Venn diagram of RBP genes associated with super-enhancers across different human smooth muscle tissues. Skeletal muscle was used as an outlier. RBPs common to all smooth muscle tissues but not skeletal muscle are shown on the left. (**C**) Schematic of the AS event determining the two major RBPMS isoforms, RBPMS A (red) and RBPMS B (yellow). (**D**) RT-PCR analysis of SMC splicing markers, *Actn1* and *Tpm1*, in differentiated (D) and proliferative (P) PAC1 cells. Schematic of the regulated mutually exclusive splicing events on top and respective isoforms products on the left. Values shown are the quantified PSI (percent spliced in) of the smooth muscle isoforms (SM) ± standard deviation (*n* = 3). (**E**) qRT-PCR analysis of *Rbpms* (all isoforms), *Rbpms2* and SMC differentiation markers *Acta2*, *Cnn1* and *Smtn*, in PAC1 cells D (green) and P (blue). Expression was normalized to the average of two housekeepers (*Gapdh* and *Rpl32*) and the mean of the relative expression is shown (*n* = 3). Each point shows data from an individual sample. Statistical significance was performed using Student's t-test (*p<0.05, **p<0.01, ***p<0.001). (**F**) Western blots for RBPMS in D and P PAC1 cells. ACTA2 is a SMC differentiation marker and GAPDH a loading control. A and B indicates the two RBPMS isoforms. (**G**) Immunofluorescence in D and P PAC1 cells for RBPMS. DAPI staining for nuclei.

The online version of this article includes the following figure supplement(s) for figure 1:

**Figure supplement 1.** RBPMS is highly expressed in SM tissues.
**Figure supplement 2.** mRNA abundance analysis of the rat aorta SMC de-differentiation RNA-Seq.
**Figure supplement 3.** Alternative splicing analysis of the rat aorta SMC de-differentiation RNA-Seq.

Here, we used the approach suggested by Jangi and Sharp, to identify candidate AS master regulators as RBP-encoding genes whose cell-specific expression is driven by super-enhancers (*Jangi and Sharp, 2014*). We identified R̲NA-B̲inding P̲rotein with M̲ultiple S̲plicing (RBPMS), a protein not previously known to regulate splicing, as a critical regulator of numerous AS events in SMCs. RBPMS is highly expressed in differentiated SMCs, where it promotes AS of genes that are important for SMC function. These include many components of the actin cytoskeleton and focal adhesion machineries, modulation of whose function is key to the transition from contractile to motile phenotypes. RBPMS also targets other splicing regulators, post-transcriptional regulators and the key SMC transcription factor Myocardin where RBPMS promotes inclusion of an exon that is essential for maximal SMC-specific activity. RBPMS therefore meets many of the criteria expected of an AS master regulator in SMCs, and its identification validates the approach of identifying key cell-specific regulators via the super-enhancer-proximity of their genes.

## Results

### RBPMS is a SMC splicing regulator

To identify potential SMC master splicing factors we used a catalog of 1542 human RBPs (*Gerstberger et al., 2014*) and data-sets of super-enhancers from three human SMC-rich tissues (aorta, bladder, stomach smooth muscle) and skeletal muscle (*Hnisz et al., 2013*) (*Supplementary file 1*). Nine RBP genes were associated with super-enhancers in all SMC tissues but not skeletal muscle (*Figure 1B*). Using the dbSUPER database (*Khan and Zhang, 2016*), which uses more stringent distance constraints for associating genes with superenhancers (see Materials and methods), we identified two candidates, of which only RBPMS was shared with our original nine candidates. Examination of *RBPMS* expression in human tissues from the Genotype-Tissue Expression (GTEX) project (*GTEx Consortium, 2013*) showed the top eight expressing tissues to be SMC rich, including three arteries (*Figure 1—figure supplement 1A*). RNA-Seq data from rat aorta SMCs showed that *Rbpms* levels decreased 3.8 fold during de-differentiation from tissue to passage 9 cell culture (*Figure 1—figure supplement 1A–C*), in parallel with known SMC transcriptional markers (*Acta2, Cnn1, Smtn*, (*Figure 1—figure supplement 1B,C*) and AS events (*Tpm1* and *Actn1*, *Figure 1—figure supplement 3A,C*). Moreover, of the starting candidate RBPs (*Figure 1B*) *Rbpms* was the most highly expressed of those that were down-regulated from tissue to culture (*Figure 1—figure supplement 2B,C* green labels). Expression of the *Rbpms2* paralog also decreased upon de-differentiation, but its absolute level of expression was >10 fold lower than *Rbpms* (*Figure 1—figure supplement 2B,C*). Other RBPs implicated in AS regulation in SMCs (PTBP1, MBNL1, QKI) either showed more modest changes or higher expression in de-differentiated cells (*Figure 1—figure supplement 2B,C*). In the PAC1 rat SMC line *Rbpms* mRNA levels decreased by ~10 fold between differentiated and proliferative states in parallel with SMC marker AS events (*Figure 1D*) and genes (*Figure 1E*). *Rbpms2* was expressed at much lower levels than *Rbpms*, and did not alter expression between PAC1 cell states (*Figure 1E*). RBPMS protein decreased to undetectable levels in proliferative PAC1 cells in parallel with smooth muscle actin (ACTA2) (*Figure 1F*). Immunofluorescence microscopy also showed higher levels of RBPMS in differentiated PAC1 cells where it was predominantly nuclear (*Figure 1G*), consistent with the hypothesis that it regulates splicing.

To further investigate RBPMS expression we cloned cDNAs from PAC1 cells, representing seven distinct mRNA isoforms. These encoded two major protein isoforms (RBPMSA and RBPMSB) sharing a common N-terminus and RNA Recognition Motif (RRM) domain. RBPMSA and B differed by short C-terminal tails encoded by alternative 3′ end exon 7 and exon 8 (*Figure 1C*), and corresponded in size to the two protein bands seen in western blots (*Figure 1F*). Other mRNA isoforms differed by inclusion or skipping of exon six and by alternative 3′ UTR exon inclusion. RBPMS and RBPMS2, which are 70% identical, have a single RRM domain that is responsible for both RNA binding and dimerization (*Sagnol et al., 2014*; *Teplova et al., 2016*). Optimal RBPMS -binding sites consist of tandem CACs separated by a spacer of ~1–12 nt (*Farazi et al., 2014*; *Soufari and Mackereth, 2017*). We found significant enrichment of $CACN_{1-12}CAC$ motifs within and upstream of exons that are less included in differentiated compared to cultured rat aorta SMCs (*Figure 1—figure supplement 3D*). These are locations at which many splicing regulators mediate exon skipping (*Fu and Ares, 2014*; *Witten and Ule, 2011*). Consistent with this, the SMC-specific mutually exclusive exon pairs in *Actn1* and *Tpm1* (*Gooding and Smith, 2008*; *Southby et al., 1999*) both have conserved clusters of CAC motifs upstream of the exon that is skipped in differentiated SMCs (see below).

In summary, the presence of super-enhancers at the *RBPMS* gene in SMC-rich tissues, its wide dynamic range of expression between differentiated SMCs and other tissues and proliferative SMCs (*Figure 1*, *Figure 1—figure supplements 1–3*), the nuclear localization of RBPMS, and the presence of potential RBPMS-binding sites adjacent to known SMC-regulated exons are all consistent with the hypothesis that RBPMS might act as a master regulator of AS in differentiated SMCs.

### RBPMS promotes a differentiated SMC splicing program

To investigate the roles of RBPMS in shaping SMC transcriptomes we manipulated levels of RBPMS expression in differentiated and proliferative PAC1 cells (*Figure 2A,B*). We used siRNAs to knock-down all *Rbpms* isoforms in differentiated PAC1 cells, achieving ~75% depletion (*Figure 2B*). In parallel, proliferative PAC1 cells were transduced with pINDUCER lentiviral vectors (*Meerbrey et al.,*

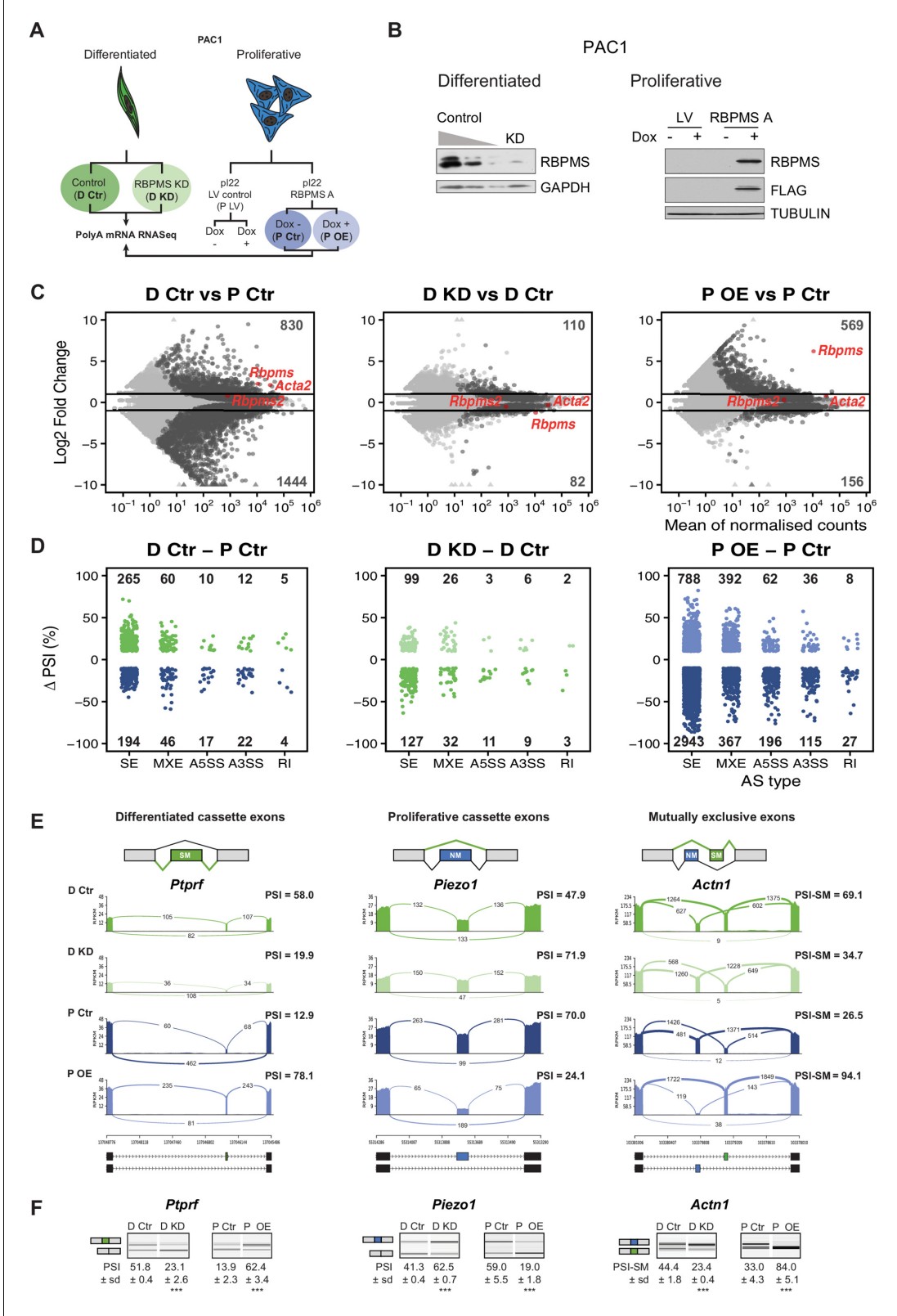

**Figure 2.** RBPMS regulates AS in PAC1 cells. (**A**) Schematic of experimental design of RBPMS knockdown and overexpression in PAC1 cells. (**B**) Western blots for RBPMS in PAC1 knockdown, left, and inducible lentiviral overexpression, right. FLAG antibodies were also used for the overexpression of 3xFLAG tagged RBPMS. GAPDH and TUBULIN were used as loading controls. (**C**) MA plots of alterations in mRNA abundance in PAC1 dedifferentiation, left, RBPMS knockdown, middle, and RBPMS A overexpression, right. Dark gray: genes with significant changes (p-adj <0.05). *Figure 2 continued on next page*

*Figure 2 continued*

Light gray: genes with p-adj ≥0.05. Red: *Rbpms*, *Rbpms2* and the SMC marker, *Acta2*. Numbers of up and down-regulated are shown at top and bottom. Horizontal lines; log2 fold change = 1 and −1. (**D**) AS changes (FDR < 0.05 and ΔPSI greater than 10%) in PAC1 cell dedifferentiation, left, RBPMS knockdown, middle, and RBPMS A overexpression, right. ASE were classified into skipped exon (SE), mutually exclusive exon (MXE), alternative 5′ and 3′ splice site (A5SS and A3SS) and retained intron (RI) by rMATS. Numbers indicate the number of significant ASE of each event type between the conditions compared. (**E**) Sashimi plots of selected ASEs. *Ptprf* is shown as a differentiated cassette exon (green), *Piezo1* as a proliferative cassette exons (blue) and *Actn1* as a MXE. The numbers on the arches indicate the number of reads mapping to the exon-exon junctions. PSI values for the ASE are indicated for each condition. Values correspond to the mean PSI calculated by rMATS. In the case of the *Actn1* MXE, the percent inclusion of the SM exon is shown (PSI-SM). Schematic of the mRNA isoforms generated by the alternative splicing are found at the bottom as well as the chromosome coordinates. (**F**) RT-PCR validation of ASEs from panel E. Values shown are the mean of the PSI ± standard deviation (*n* = 3). Statistical significance was calculated using Student's t-test (*p<0.05, **p<0.01, ***p<0.001). See *Figure 2—figure supplement 3* for more ASEs validated in the RBPMS knockdown and RBPMS A overexpression by RT-PCR.

The online version of this article includes the following figure supplement(s) for figure 2:

**Figure supplement 1.** Principal Component Analysis of PAC1 RNAseq samples and overlap of genes regulated at the splicing and abundance levels.

**Figure supplement 2.** AS changes of SM genes are specific to RBPMS expression.

**Figure supplement 3.** ΔPSI values determined by RT-PCR and RNA-Seq show good agreement.

*2011*) to allow Doxycycline inducible over-expression of RBPMSA. No basal RBPMS expression was observed in proliferative cells, but upon induction substantial expression was observed from the FLAG-RBPMSA vector but not from the empty lentiviral vector (LV) (*Figure 2B*). The effects of manipulating RBP levels can sometimes be compensated by related family members (*Mockenhaupt and Makeyev, 2015*). However, *Rbpms2* levels were not affected by any of the treatments and *Rbpms2* knockdown, either alone or in combination with *Rbpms*, had no effects upon tested AS events (data not shown).

RNA samples from *Rbpms* knockdown and overexpression experiments were prepared for Illumina poly(A) RNAseq. Data was analyzed for changes in mRNA abundance using DESEq2 (*Love et al., 2014*) (*Figure 2C*, *Supplementary file 2*), and for changes in AS using rMATS (*Shen et al., 2014*) (*Figure 2D*, *Supplementary file 3*). In addition to analysis of effects of RBPMS depletion and overexpression, comparison of the differentiated and proliferative PAC1 control samples revealed changes associated with differentiation state. Principal Component Analysis based on mRNA abundance showed clear separation of differentiated and proliferative samples (PC1, 82% variance, *Figure 2—figure supplement 1A*). This suggests that at the level of mRNA abundance the differences between differentiated and proliferative PAC1 cells far outweigh any effects of manipulating RBPMS levels. Consistent with this, RBPMS knockdown was associated with ~10 fold fewer changes at the transcript abundance level (110 increased, 82 decreased, padj <0.05, fold change >2 fold) compared to the differentiated vs proliferative control comparison (830 increased, 1444 decreased, *Figure 2C*). RBPMS overexpression led to an intermediate number of changes in mRNA abundance levels, but only 29 genes were affected by both overexpression and knockdown, and of these only four genes other than *Rbpms* were regulated reciprocally.

RBPMS knockdown and overexpression had substantial effects at the level of AS affecting all types of AS event (*Figure 2D*, *Figure 2—figure supplement 1B*). RBPMS knockdown led to changes in 318 AS events (FDR < 0.05, |ΔPSI| > 0.1, where PSI is Percent Spliced In), which was only 2-fold less than the number of events regulated between control differentiated and proliferative cells. Cassette exons were the largest group of events, with roughly equal numbers of up- and down-regulated exons (*Figure 2D*). RBPMS overexpression led to a larger number of AS changes (4934 regulated events), probably resulting from the combination of RBPMS expression in excess of levels usually present in differentiated PAC1 cells (*Figure 2C*) and also because we expressed the more active RBPMSA isoform (see below). Cassette exons affected by RBPMS overexpression were strongly skewed (80%) towards greater exon skipping. A subset of ASEs observed in the RNA-Seq experiments, encompassing RBPMS activated and repressed cassette exons and mutually exclusive exons were validated by RT-PCR (*Figure 2E,F*, *Figure 2—figure supplements 2* and *3*). ΔPSI values determined by RT-PCR and RNA-Seq were in good agreement (*Figure 2—figure supplement 3B*). As an additional negative control, doxycycline induction of empty lentiviral transduced cells had no effect on RBPMS-regulated AS events (*Figure 2—figure supplement 2A*).

In each of the three comparisons, the overlap between genes regulated at the levels of splicing and mRNA abundance was small: 0.8% of all genes regulated at splicing or abundance level for RBPMS knockdown, 3.2% for RBPMS overexpression, and 1.6% for PAC1 phenotype, (*Figure 2—figure supplement 1C*). However, there were substantial overlaps of ASEs regulated by RBPMS knockdown, overexpression and PAC1 phenotype (*Figure 3A*). Twenty percent of ASEs regulated in PAC1 cell differentiation were congruently regulated by RBPMS knockdown, representing 40% of ASEs affected by RBPMS knockdown (*Figure 3A and B* left panel). The high correlation ($R^2 = 0.95$) suggests that for these 127 events changes in RBPMS expression are sufficient to explain their differentiation-specific splicing changes. Similarly, RBPMS overexpression shared 180 events in common with PAC1 differentiation status (28% of differentiation specific events, 3.7% of overexpression regulated events). The ΔPSI correlation for these AS events was lower than for RBPMS knockdown ($R^2 = 0.86$, *Figure 3A and B* right panel). Sixty seven events regulated by RBPMS knockdown were reciprocally regulated by RBPMS overexpression, of which 52 were shared across all the three comparisons (8.2% of differentiation specific events), as exemplified by events in *Ptprf*, *Piezo1* and *Actn1* (*Figure 2E*). Hierarchical clustering of cassette exons regulated between PAC1 phenotypes across all 12 samples also revealed two clusters of RBPMS-responsive events where knockdown and overexpression were sufficient to reciprocally convert the splicing pattern to that of the other cellular phenotype (*Figure 3E*, clusters 1 and 4, containing RBPMS activated and repressed exons respectively).

Many AS events in differentiated PAC1 cells do not reach the fully differentiated splicing pattern characteristic of tissue SMCs. However, for some events overexpression of RBPMS led to splicing patterns similar to tissue SMCs. For example the *Actn1* SM exon is included to 69% in differentiated PAC1 cells, but to 93–94% in RBPMS overexpressing cells (*Figure 2E*) and aorta tissue (*Figure 1—figure supplement 3C*). We therefore hypothesized that some ASEs regulated by RBPMS overexpression but not knockdown, might reflect tissue SMC AS patterns that are not usually observed in cultured SMCs. To address this possibility, we used RNA-Seq data monitoring de-differentiation of rat aorta SMCs (*Figure 1—figure supplements 2* and *3*). Of 1714 ASES regulated between tissue and passage 9 SMCs, 265 (15%) were also regulated by RBPMS overexpression in PAC1 cells (*Figure 3C,D*, $R^2 = 0.68$). Strikingly, hierarchical clustering of cassette exons regulated between aorta tissue and passage nine cultured SMCs showed that the RBPMS overexpression sample clustered together with tissue, away from all other samples (*Figure 3F*). Two clusters of AS events shared very similar splicing patterns in tissue and RBPMSA overexpression, and differed in all remaining samples (*Figure 3F*). Cluster 1 comprised 28 RBPMS-activated exons, of which 20 were not regulated between PAC1 phenotypes. For example, inclusion of a cassette exon in *Fermt2* was observed only in tissue and RBPMS overexpression samples (*Figure 3G*). Similarly, inclusion of *Cald1* exon four in conjunction with a downstream 5′ splice site on exon 3a, producing the hCald1 marker in tissue SMCs (*Figure 3G*, *Figure 3—figure supplement 1C,D*), and inclusion of the meta-vinculin exon (*Figure 3—figure supplement 1A*) were also only seen upon RBPMS overexpression. Cluster four contained exons that are skipped in tissue and upon RBPMS overexpression (e.g. *Tsc2*, *Figure 3G*), the majority of which are not regulated in PAC1 differentiated cells. Likewise *Tpm1* mutually exclusive exon three is skipped nearly completely upon RBPMS overexpression in a similar pattern to tissue (*Figure 3—figure supplement 1B*). Upregulation of RBPMS expression therefore appears to be sufficient to promote a subset of splicing patterns usually only observed in differentiated SMCs in vivo.

## RBPMS directly regulates exons with associated CAC motifs

To address whether RBPMS directly regulates target exons we looked for enrichment of its binding motif (*Farazi et al., 2014*) adjacent to cassette exons regulated by RBPMS knockdown or overexpression. $CACN_{1-12}CAC$ motifs, the optimal binding motif for RBPMS dimers, were significantly enriched around exons that were activated or repressed by RBPMS, with a similar position-dependent activity as other splicing regulators (*Figure 4A*). Exons repressed by RBPMS showed strong enrichment of motifs within the exon and the immediate ~80 nt upstream intron flank, while exons activated by RBPMS showed motif enrichment within the downstream intron flank. Consistent with the contribution of RBPMS to the AS changes between PAC1 differentiation states, $CACN_{1-12}CAC$ motifs were also enriched upstream of and within exons that are more skipped in differentiated cells, and downstream of exons that are more included in differentiated cells (*Figure 4A*). Moreover, in the set of exons activated by RBPMS overexpression, $CACN_{1-12}CAC$ motifs were not only enriched

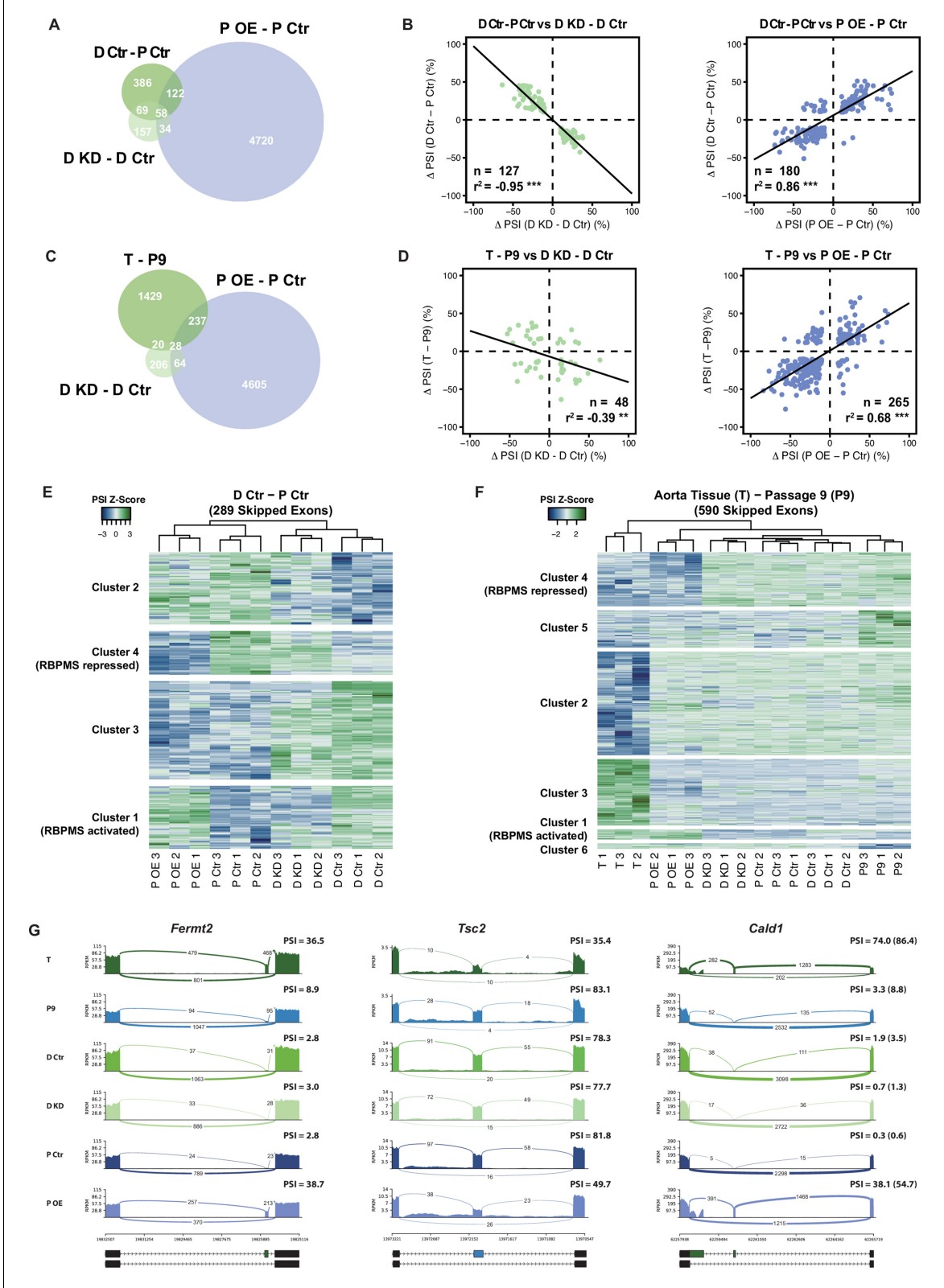

**Figure 3.** RBPMS recapitulates the AS program of differentiated PAC1 and aorta tissue. (**A**) Venn diagram of significant ASEs (FDR < 0.05 and ΔPSI cutoff of 10%) identified in the PAC1 dedifferentiation, RBPMS knockdown and RBPMS A overexpression comparisons. (**B**) ΔPSI correlation of overlapping ASEs (FDR < 0.05 and ΔPSI cutoff of 10%) from the PAC1 cells dedifferentiation and RBPMS knockdown, left, or RBPMS A overexpression, right. The line indicates the linear regression model. Statistical significance was carried out by Pearson correlation test. n is the number of ASEs

*Figure 3 continued on next page*

Figure 3 continued

assessed, r² is the correlation coefficient. Statistical significance of the correlation: *p<0.05, **p<0.01, ***p<0.001. (C) Venn diagram of significant ASEs identified in RBPMS knockdown, overexpression and aorta tissue to passage nine comparisons. (D) ΔPSI correlation of overlapping ASEs from the rat aorta tissue dedifferentiation and RBPMS knockdown, left, or RBPMS A overexpression, right. (E) Heatmap of 289 SEs regulated in the PAC1 dedifferentiation comparison (D Ctr - P Ctr). Each column represents a replicate sample (1-3) from RBPMS knockdown (D Ctr and D KD) or overexpression (P Ctr and P OE). Rows are significant SE events from PAC1 dedifferentiation. Rows and columns were grouped by hierarchical clustering. Blue and green colors in the Z-score scaled rows represent low and high PSI values respectively. (F) Heatmap of 590 SEs regulated in the rat aorta tissue dedifferentiation comparison (T - P9). Each column represents one sample from either aorta tissue dedifferentiation (T and P9), RBPMS knockdown (D Ctr and D KD) or RBPMS overexpression (P Ctr and P OE). Rows are significant SE events from the T - P9 comparison. Rows and columns were grouped by hierarchical clustering. Blue and green colors in the Z-score scaled rows represent low and high PSI values respectively. (G) Sashimi plots of selected ASEs highly regulated in the aorta tissue and RBPMS overexpression. *Fermt2* is an RBPMS activated exon from Cluster one in panel F. *Tsc2* is an RBPMS repressed exon from Cluster 4 of panel E. *Cald1* has an RBPMS activated exon 4 and downstream five splice site on exon 3b. For Cald1, a manually calculated PSI in parentheses takes into account the A5SS which was not included in the rMATS annotation.

The online version of this article includes the following figure supplement(s) for figure 3:

**Figure supplement 1.** RBPMS recapitulates the aorta tissue AS pattern of *Vcl*, *Tpm1* and *Cald1*.

downstream, but also significantly depleted in the repressive locations within and upstream of the exon (*Figure 4A,B*). This suggests that binding in repressive locations might be dominant over activation.

To test whether RBPMS regulates AS events by directly binding to CAC motifs we co-transfected HEK293 cells with RBPMS expression vectors and minigenes of representative activated (*Flnb*) and repressed (*Tpm1*, *Actn1*) exons, with potential RBPMS-binding sites in expected locations for activation or repression (*Figure 4—figure supplements 1B–C* and *2B*). We initially established that transient expression of RBPMSA in HEK293 cells was sufficient to switch AS of endogenous *FLNB*, *TPM1*, *MPRIP* and *ACTN1* towards the SMC splicing pattern (*Figure 4C,G* and *Figure 4—figure supplement 1A*). For comparison, we also transfected expression constructs for the RBPMSB isoform and the paralog RBPMS2. We found that RBPMSB had lower activity for some events (*ACTN1*, *Figure 4—figure supplement 1A*), but that transfected RBPMS2 had similar activity to RBPMSA in all cases. RBPMSA and RBPMS2 also strongly activated inclusion of the *Flnb* H1 exon in a minigene context while RBPMSB activated to a lower extent (*Figure 4D*). *Tpm1* exon three is the regulated member of a pair of mutually exclusive exons and is repressed in SMCs (*Ellis et al., 2004*; *Gooding et al., 1994*). MBNL and PTBP proteins promote this repression but are not sufficient to switch splicing (*Gooding et al., 2013*; *Gooding et al., 1998*). In contrast, RBPMSA expression was sufficient to cause a near complete switch from exon inclusion to skipping (*Figure 4H*). RBPMS2 had lower activity, but RBPMSB was by far the least active protein. Likewise, RBPMSA and RBPMS2 completely switched splicing of *Actn1* constructs (*Gromak et al., 2003*; *Southby et al., 1999*) from the NM to the SM mutually exclusive exon, while RBPMSB was nearly inactive (*Figure 4—figure supplement 2A*). A construct containing only the *Actn1* SM exon was unresponsive to cotransfection, while a construct containing only the NM exon, which has three upstream CAC clusters (*Figure 4—figure supplement 1B*) showed a complete switch from inclusion to skipping upon cotransfection of RBPMSA or RBPMS2, but not RBPMSB. Thus, for two mutually exclusive events RBPMSA is able to switch splicing to the SMC pattern by repressing the exon that is usually used in non-SMCs, while RBPMSB is less active.

We mutated CAC motifs in suspected binding sites to CCC, which disrupts RBPMS binding (*Farazi et al., 2014*). Mutation of 12 CACs downstream of the *FlnB* H1 exon had no effect on basal splicing, but the H1 exon was completely resistant to activation by RBPMSA (*Figure 4E*). Likewise, mutation of 9 CAC motifs upstream of *Tpm1* exon three had no effect on exon inclusion in the absence of RBPMSA, but completely prevented exon skipping in response to RBPMSA (*Figure 4I*, *Figure 4—figure supplement 2B*), while mutations of individual clusters had intermediate effects (*Figure 4—figure supplement 2B*). The response of both *Flnb* H1 exon and *Tpm1* exon three therefore depends on nearby CAC motifs. To test whether these are binding sites for RBPMS, we used in vitro transcribed RNAs and recombinant RBPMSA and B (*Figure 4F,J* and *Figure 4—figure supplement 4*). Using both electrophoretic mobility shift assay (EMSA) and UV crosslinking, both RBPMSA and B were found to bind to the *FlnB* wild type RNA (apparent Kd ~0.5 μM), but not to the mutant RNA (*Figure 4F*). With *Tpm1*, RBPMSA and B bound to the WT RNA as indicated by EMSA assays

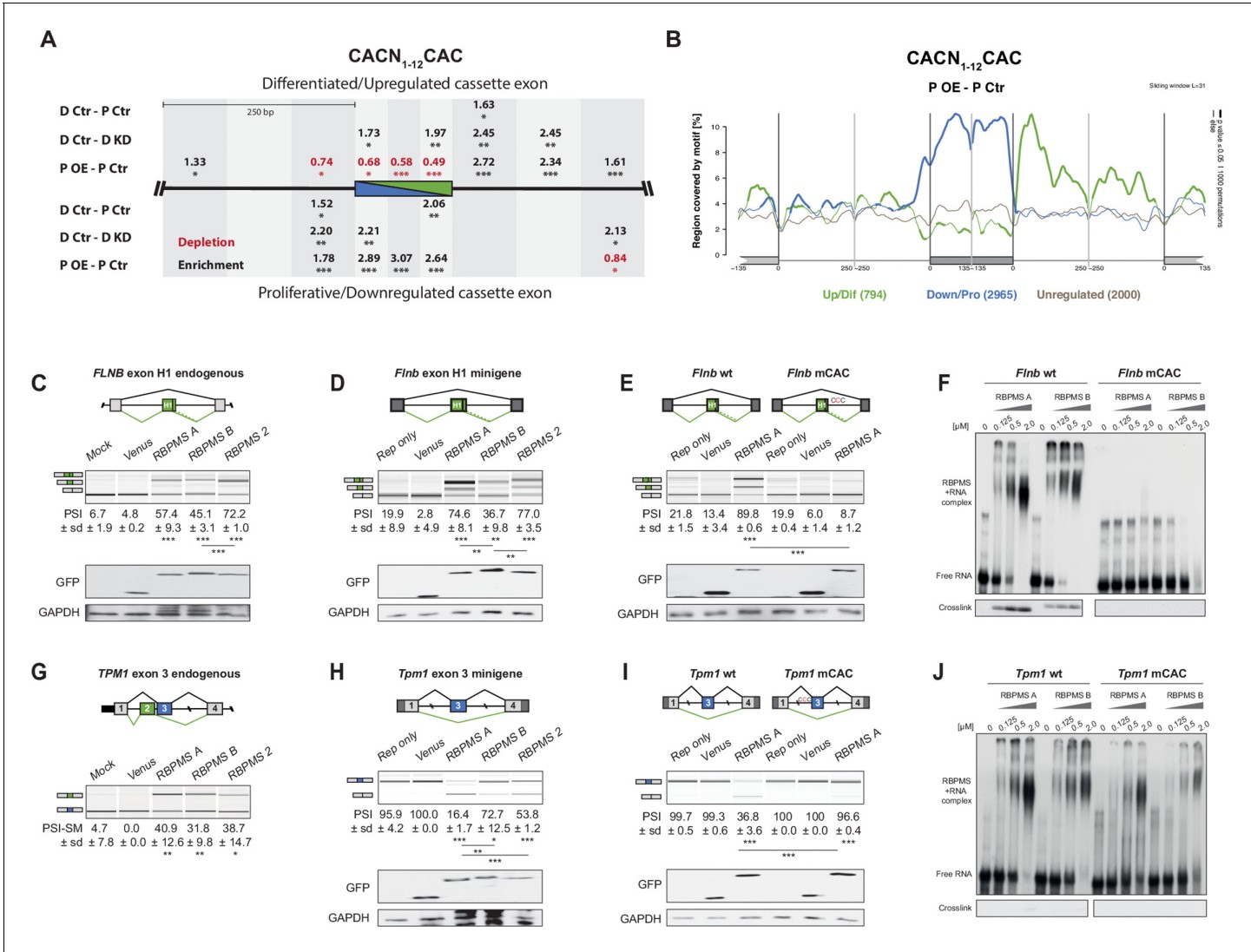

**Figure 4.** RBPMS directly regulates exons associated with CAC motifs. (**A**) RBPMS motif ($CACN_{1-12}CAC$) enrichment in differentially alternatively spliced cassette exons in the PAC1 dedifferentiation (D Ctr - P Ctr), RBPMS knockdown (D Ctr - D KD) and RBPMS overexpression (P OE - P Ctr) comparisons. Values indicate degree of motif enrichment (black) or depletion (red). Statistical significance: *p<0.05, **p<0.01,***p<0.001. (**B**) RBPMS motif ($CACN_{1-12}CAC$) map from RBPMS overexpression (P OE - P Ctr). Upregulated (green), downregulated (blue) and unregulated (gray) cassette exons are shown. Statistical significance: p<0.05, 1000 permutations is indicated by thicker line width. (**C**) RBPMS overexpression in HEK293 cells and RT-PCR of endogenous *FLNB* exon H1 splicing. Schematic of *Flnb* exon H1 as a differentiated cassette exon activated by RBPMS. The splicing pattern of *FLNB* was tested upon overexpression of Venus tagged RBPMS A, B and 2. A mock and a Venus control were also tested in parallel. (**D**) Effect of RBPMS overexpression on rat *Flnb* splicing reporter in HEK293 cells. Schematic of the *Flnb* reporter and RT-PCR for the splicing patterns of *Flnb* reporter upon coexpression of RBPMS isoforms and RBPMS2. (**E**) *Flnb* reporter with point mutations disrupting RBPMS motifs (CAC to CCC) were tested for RBPMS A regulation. Left, wild-type reporter and right, mutant CAC reporter (mCAC) RT-PCRs. (**F**) Electric mobility shift assay (EMSA) for in vitro binding of recombinant RBPMS A and B to in vitro transcribed wild-type and mCAC RNAs of *Flnb* exon H1 downstream intron sequence (top left and right). Lower panel: UV-crosslinking of same samples. (**G**) RBPMS overexpression in HEK293 cells and RT-PCR of endogenous *TPM1* MXE exon 2 and 3 splicing. The splicing pattern of *TPM1* was tested upon overexpression of Venus tagged RBPMS A, B and 2. A mock and a Venus control were also tested in parallel. (**H**) Effect of RBPMS overexpression on rat *Tpm1* splicing reporter in HEK293 cells. Schematic of the *Tpm1* reporter and RT-PCR for the splicing patterns of *Tpm1* reporter upon RBPMS isoforms and paralog. (**I**) *Tpm1* reporter with point mutations disrupting RBPMS motifs (CAC to CCC) was tested for RBPMS A regulation. Left, wild-type reporter and right, mutant CAC reporter (mCAC) RT-PCRs. (**J**) In vitro binding of recombinant RBPMS A and B to in vitro transcribed wild-type and mCAC RNAs of *Tpm1* exon three upstream intron sequence. EMSA, top and UV crosslinking, bottom. In (**F**) and (**J**), in vitro binding assays were carried out using recombinant protein in a serial dilution (1:4) in a range of 0.125 to 2 μM. In panels (**C–E**) and (**G–I**), values are the mean PSI ± sd (*n* = 3). For *Flnb* cassette exon, the PSI is the sum of both short and long isoforms generated by a A5SS event. For *Tpm1* MXE, the SM exon PSI is shown (PSI-SM). Schematics of the splicing isoforms indicate the PCR products, differentiated (green) and proliferative (blue). Statistical significance was calculated using Student's t-test (*p<0.05, **p<0.01, ***p<0.001). Western blot anti-GFP and GAPDH, loading control, were carried out

*Figure 4 continued on next page*

*Figure 4 continued*

to assess RBPMS isoform overexpression in HEK293 (panels (**C-E, H, I**). The western blot in (**C**) is also a representative of (**G**), since both RT-PCRs (*FLNB* and *TPM1*) are from the same overexpression experiment.

The online version of this article includes the following figure supplement(s) for figure 4:

**Figure supplement 1.** RBPMS promotes the SM splicing of *MPRIP* and *ACTN1* in HEK293 cells.
**Figure supplement 2.** RBPMS represses the splicing of the NM exon of *Actn1* and *Tpm1*.
**Figure supplement 3.** RBPMS splicing activity requires RNA binding and dimerization.
**Figure supplement 4.** RBPMS recombinant proteins.

(Kd ~0.5 μM) (*Figure 4J*). Binding was reduced by the mutations that abrogated RBPMSA repression of *Tpm1* exon 3 (Kd >0.5 μM). Finally, we tested the effects of mutations in RBPMS to disrupt the RNA binding and dimerization interfaces of the RRM (*Sagnol et al., 2014*; *Teplova et al., 2016*). Both RNA binding and dimerization mutants of RBPMSA had lost all splicing regulatory activity upon overexpression in HEK293 cells (*Figure 4—figure supplement 3*), revealing RBPMS dimer-dependent splicing activity. These data therefore show that RBPMS can inhibit splicing by binding to CAC clusters upstream of exons (*Tpm1*) and activate splicing by downstream binding (*Flnb*).

## RBPMS-regulated splicing targets the cytoskeleton and cell adhesion

To investigate the functional importance of RBPMS-regulated AS, we carried out Gene Ontology (GO) analysis of the genes whose splicing or expression levels were regulated by RBPMS (*Figure 5A*, *Figure 5—figure supplement 1A*,*Supplementary files 4* and *5*), or between SMC differentiation states (*Figure 5—figure supplement 1B,C*). Splicing events affected by RBPMS-knockdown affected genes involved in processes, components and functions important for SMC biology, such as cytoskeleton, cell projection, cell junction organization and GTPase regulation (*Figure 5A*). These categories were very similar to those affected by AS during PAC1 cell dedifferentiation (*Figure 5—figure supplement 1B*). Events regulated by RBPMS knockdown were also enriched within genes that are associated with super-enhancers in SMC tissues (*Figure 5B*) and therefore inferred to be important for SMC identity. In contrast, the relevance of GO terms associated with RBPMSA overexpression to SMC biology was less clear (*Figure 5—figure supplement 1A*), despite the presence of many events that were reciprocally regulated by RBPMS knockdown. RBPMS overexpression AS targets were also not enriched for aorta super-enhancer-associated genes (p=0.37). This discrepancy might be accounted for by the high level of RBPMS overexpression leading to changes in some AS events that are not physiological targets. At the RNA abundance level enriched GO terms associated with RBPMS knockdown or overexpression did not align with those associated with differentiation, mainly being associated with stress responses (*Supplementary file 5*).

To further explore the consequences of RBPMS regulation of AS we carried out network analysis using STRING (*Szklarczyk et al., 2017*) (*Figure 5C*). To ensure that all target events were biologically relevant we used only AS events that are coregulated by RBPMS knockdown and PAC1 differentiation state (*Figure 3A,E*) or by RBPMS overexpression and in aorta tissue (*Figure 3C,F*). We also restricted the output network to high confidence interactions. RBPMS targets comprised a network (*Figure 5C*) focused on functions associated with cell-substrate adhesion (yellow nodes) and the actin cytoskeleton (blue). Six of the network proteins (SORBS1, CALD1, PDLIM5, PDLIM7, ACTN1 and ARHGEF7) are also components of the consensus integrin adhesome (*Horton et al., 2015*), which mechanically connects, and mediates signalling between, the actin cytoskeleton and the extracellular matrix. Underlining the importance of this network to SMC function, many of the genes are themselves super-enhancer-associated in one or more smooth muscle tissues (bold text, *Figure 5C*). Actomyosin activity, and its mechanical connection to the extracellular matrix are important for both the contractile and motile states of SMCs (*Min et al., 2012*). It appears that RBPMS plays an important role in modulating the activity of this protein network to suit the needs of the two cell states. Despite these coordinated changes in AS, we were unable to observe obvious changes in PAC1 cell phenotype after under the overexpression and knockdown treatments used for RNA analysis. We reasoned that more sustained perturbation of RBPMS levels might be required to allow turnover of protein isoforms before phenotypes would become apparent. We therefore prolonged RBPMS knockdown to a total of 120 hr (*Figure 5D–F* and *Figure 5—figure supplements 2* and *3*). Effective

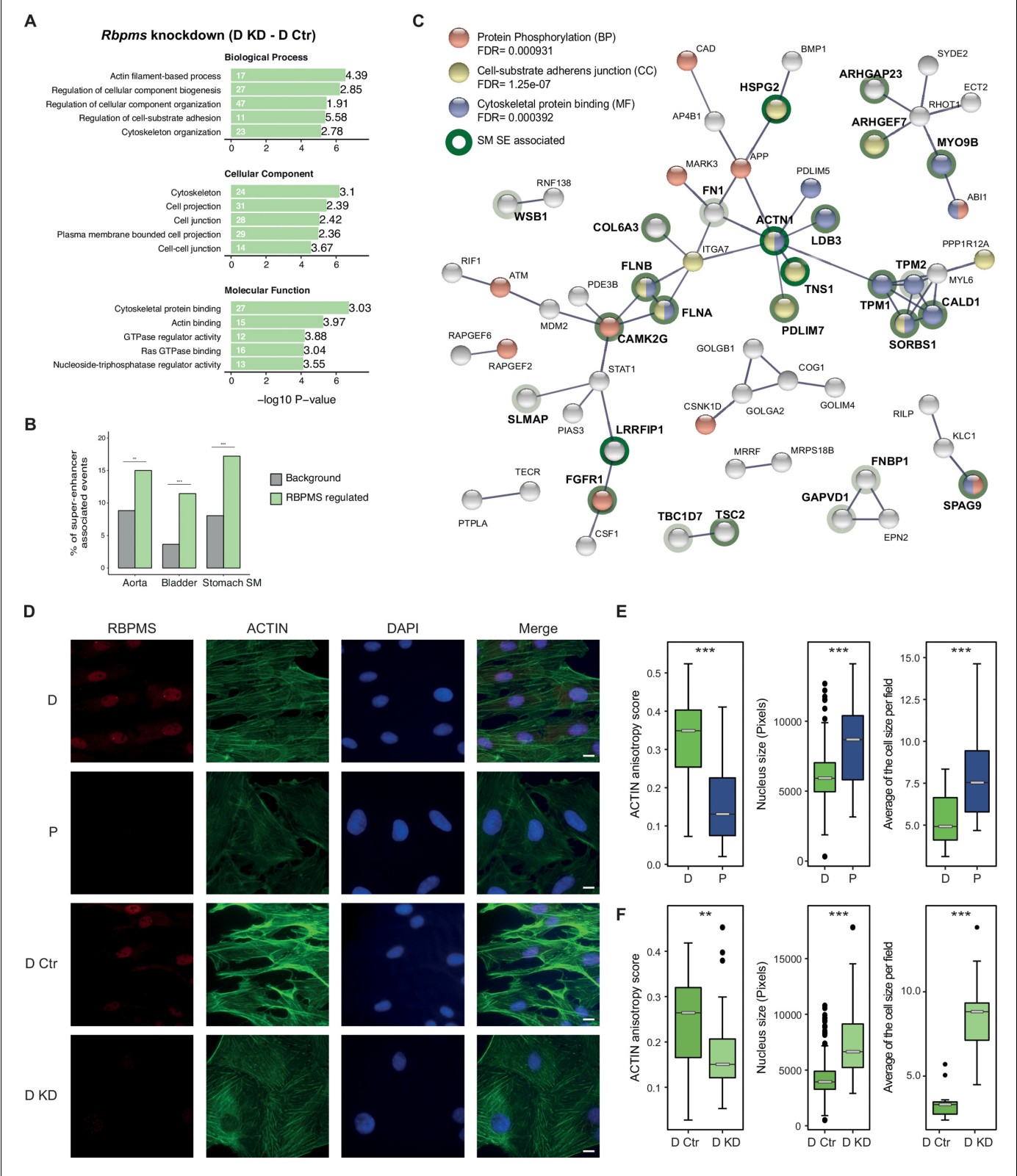

**Figure 5.** RBPMS regulates functionally important targets in SMCs. (**A**) GO analysis of genes with cassette exons regulated in RBPMS knockdown. The top five enriched GO terms in the three categories (biological process, molecular function and cellular component) are shown. Values within and in front of the bars indicate the number of genes in the enriched term and the enrichment relative to the background list. (**B**) Enrichment of exons regulated by RBPMS knockdown within genes associated with super-enhancers in smooth muscle tissues. Background set is all cassette exon events

*Figure 5 continued on next page*

*Figure 5 continued*

(regulated and unregulated) detected by rMATS in the same experiment. Significance determined by hypergeometric P-value. (**C**) PPI network of genes showing concordant splicing regulation upon RBPMS knockdown and PAC1 differentiation status, combined with genes concordantly regulated by RBPMS overexpression and in aorta tissue datasets. PPI network was generated in STRING using experiments and database as the sources of interactions. Network edges represent the interaction confidence. Enriched GO terms (BP, biological process, MF, molecular function and CC, cellular component) were also included in the analysis and are indicated in red, blue and yellow. Super-enhancer associated gene names are in bold and are highlighted gray, light green or dark green shading according to whether they were super-enhancer associated in 1, 2 or 3 SMC tissues. (**D**) Immunofluorescence of RBPMS and actin (Phalloidin) in differentiated and proliferative PAC1 cells (D and P) and upon prolonged (120 hr) RBPMS knockdown in differentiated PAC1 cells (D Ctr and D KD). DAPI staining for cell nuclei. Scale bars 10 μm. (**E**) Left, anisotropy measurement of actin fibers in PAC1 cells D (differentiated) and P (proliferative) using the FibrilTool ImageJ macro (n = 44 and 52). Middle, nucleus size measurement shown in pixels (n = 182 and 130). Right, average of the cell size quantified per field (n = 11 and 15). (**F**) Left, anisotropy measurement of actin fibers in RBPMS knockdown (D Ctr and D KD) using the FibrilTool ImageJ macro (n = 56 and 33). Middle, nucleus size measurement shown in pixels (n = 363 and 75). Right, average of the cell size quantified per field (n = 14 and 10). In (**E**) and (**F**), statistical significance was obtained from a Mann-Whitney-Wilcoxon Test (*p<0.05, **p<0.01, ***p<0.001). Data shown are from one representative experiment carried out in triplicate.

The online version of this article includes the following figure supplement(s) for figure 5:

**Figure supplement 1.** GO analysis of genes differentially spliced upon RBPMS overexpression and PAC1 and aorta dedifferentiation.

**Figure supplement 2.** Morphological changes in PAC1 cells are specific to RBPMS knockdown.

**Figure supplement 3.** Sustained RBPMS knockdown in PAC1 cells.

knockdown of RBPMS protein and RNA, and effects upon known RBPMS-regulated AS events were all confirmed (*Figure 5—figure supplement 3*). Under these conditions we observed a modest reduction in levels of smooth muscle actin (Acta2) protein and RNA (*Figure 5—figure supplement 3*), which had not been seen with shorter knockdown. Strikingly, we observed substantial changes in cell morphology and actin organization of differentiated PAC1 cells upon prolonged RBPMS deple-tion. Compared to control cells, PAC1 cells treated with RBPMS siRNA (D KD) displayed less align-ment of actin fibers, as indicated by the lower actin anisotropy score (~0.6 fold), accompanied by larger nuclear (~1.7 fold) and cell sizes (~2.6 fold) (*Figure 5D,F*). Remarkably, the RBPMS knockdown differentiated cells closely resembled the proliferative cells in these characteristics (*Figure 5D,E*). Therefore, the control of a functionally coherent set of targets that are important for SMC morphol-ogy and function is consistent with the hypothesis that RBPMS is a master regulator of AS in SMCs.

## RBPMS controls post-transcriptional regulators

Among the direct targets of master splicing regulators are expected to be events controlling the activity of other AS regulators leading to further indirect AS changes. Consistent with this, we noted that RBPMS affected splicing of *Mbnl1* and *Mbnl2*. RBPMS promoted skipping of the 36 nt and 95 nt alternative exons of *Mbnl1* (here referred to as exons 7 and 8), as indicated by both knockdown and overexpression (*Figure 6A,B*). In *Mbnl2* the 36 nt exon was fully skipped in all conditions but the 95 nt exon was repressed by RBPMS (*Figure 6C*). Consistent with the RNA-Seq data, we obtained *Mbnl1* and *Mbnl2* cDNAs from PAC1 cells that varied by inclusion of the 36 and 95 nt exons. Corresponding shifts in MBNL1, but not MBNL2, protein isoforms could be observed by western blot (*Figure 6D*). These events affect the unstructured C-termini of MBNL proteins and have been shown to affect their splicing activity (*Sznajder et al., 2016*; *Tabaglio et al., 2018*; *Tran et al., 2011*), suggesting that RBPMS might indirectly affect some AS events by modulating MBNL activity. To test this, we made expression constructs of rat MBNL1 isoforms with and without exons 7 and 8 (FL, Δ7, Δ8, Δ7Δ8) and MBNL2 with and without exon 8 (FL and Δ8), obtained as cDNAs from PAC1 cells. When transfected into HEK293 cells, full length MBNL1 and 2 caused a shift from use of the downstream to an upstream 5' splice site (5'SS) on *NCOR2* exon 47 (*Figure 6E*), an event that is dif-ferentially regulated by MBNL isoforms (*Sznajder et al., 2016*; *Tran et al., 2011*). The shorter MBNL isoforms showed lower activity in shifting towards the upstream 5'SS, although the difference was not statistically significant for the MBNL1 shorter isoforms (*Figure 6E*). In proliferative PAC1 cells, *NCOR2* exon 47 mainly uses the upstream 5'SS, and knockdown of MBNL1 and 2 caused a signifi-cant shift to the downstream 5'SS (*Figure 6F*). Overexpression of RBPMSA, also caused a small shift to the downstream 5'SS (*Figure 6F*, also detected by rMATs, ΔPSI = 13%, FDR = $2\times10^{-6}$), but had no effect when MBNL1 and MBNL2 were knocked down. These results are consistent with MBNL1

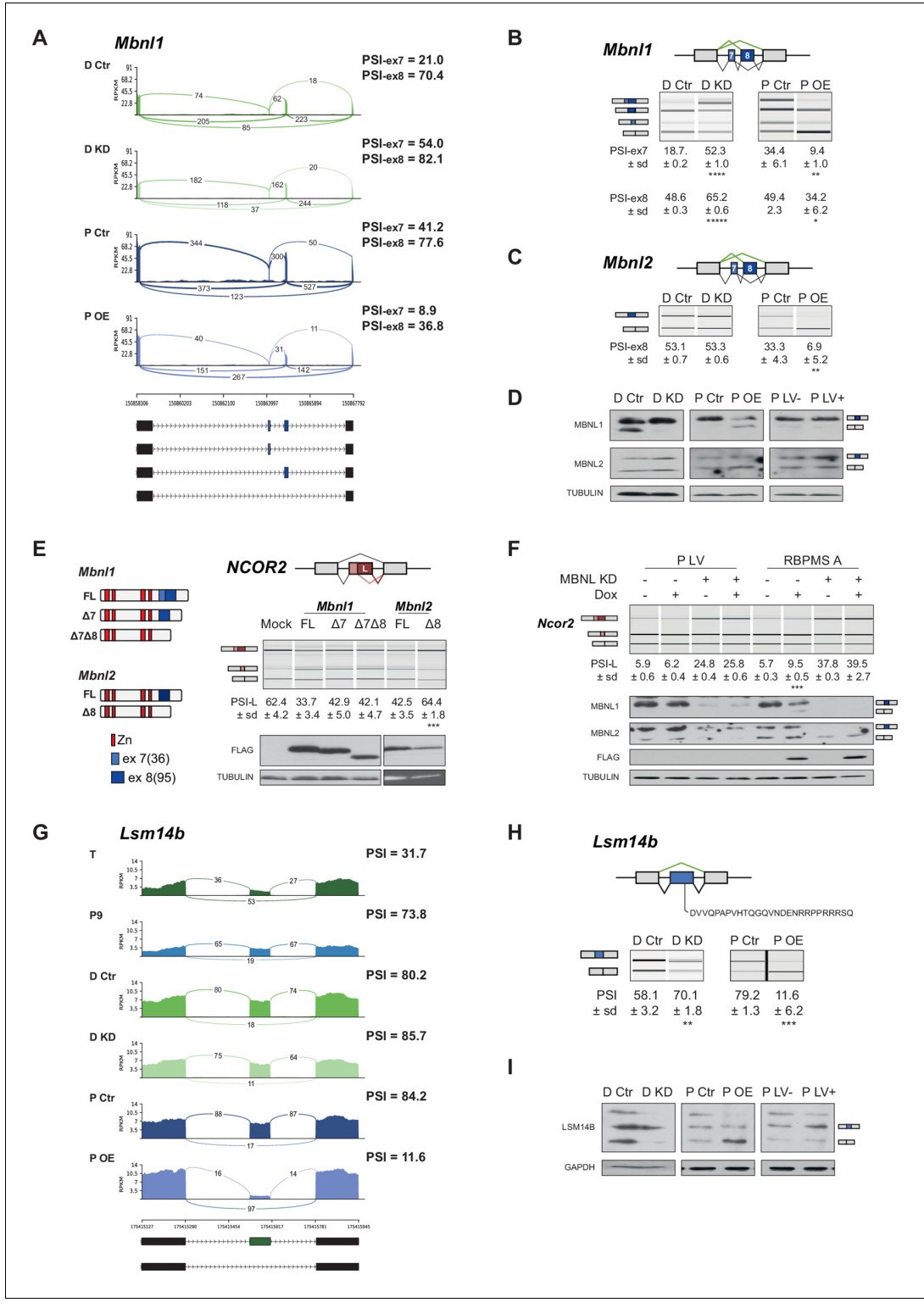

**Figure 6.** RBPMS regulates splicing of post-transcriptional regulators. (**A**) Sashimi plot of *Mbnl1* regulated exons. The numbers on top of the arches indicate the number of reads mapping to the exon-exon junctions. Mean of the PSI values calculated for each exon (exon 7–36 bp and exon 8–95 bp) are indicated for each condition (PSI-ex7 and PSI-ex8). Schematic of the different mRNA isoforms are found at the bottom. (**B**) RT-PCR validation of Mbnl1 exons 7 and 8 and (**C**) Mbnl2. Values shown are the mean of the PSI ± standard deviation (*n* = 3). The PSI values for each exon of Mbnl1 are shown. For *Mbnl2*, exon seven isoforms were not detected in the RT-PCRs. Schematics of the splicing isoforms identify the PCR products. (**D**) Western blot probing for MBNL1 and 2 in

*Figure 6 continued on next page*

*Figure 6 continued*

RBPMS knockdown and overexpression shows the MBNL1 isoform switch at the protein level. (**E**) Schematic of the different MBNL1 protein isoforms, left. Different MBNL1 and MBNL2 isoforms were overexpressed in HEK293T cells and their effect on an A5SS event in the *NCOR2* gene assessed. Schematics of *NCOR2* splicing isoforms indicates the PCR products. Western blot were probed against FLAG to verify isoform overexpression. PSI indicates use of the downstream A5SS to generate the longer isoform. (**F**) MBNL1 and 2 knockdown in inducible RBPMS A or in lentiviral control (P LV) proliferative PAC1 cells to assess the dependency of the *Ncor2* A5SS event to the MBNL1 isoform switch. Western blot for MBNL1 and 2 and FLAG to confirm MBNL knockdown and RBPMS A overexpression. In all the western blots TUBULIN was used as a loading control. Statistical significance was calculated using Student's t-test (*p<0.05, **p<0.01, ***p<0.001). (**G**) Sashimi plot of the *Lsm14b* cassette exon regulated by RBPMS. PSI values represent the mean of the PSI values calculated by rMATS analysis. (**H**) RT-PCR validation of *Lsm14b* in the RBPMS knockdown and overexpression in PAC1 cells. Schematic of the *Lsm14b* event at the top and schematic of its isoforms at the left. Peptide sequence coded by the exon is shown below the schematic. (**I**) Western blot of LSM14B in RBPMS knockdown and overexpression cells. GAPDH was used for loading control.

and 2 being direct regulators of *Ncor2* splicing, with RBPMS acting indirectly by promoting production of less active MBNL isoforms.

Another post-transcriptional regulator affected by RBPMS is LSM14B, which is involved in cytoplasmic regulation of mRNA stability and translation (*Brandmann et al., 2018*) but also shuttles to the nucleus (*Kırlı et al., 2015*). RBPMSA overexpression promoted skipping of *Lsm14b* exon 6, a pattern which is also seen in tissue SMCs (*Figure 6G,H*), and knockdown was also seen to affect this event (*Figure 6G,H*), with corresponding changes in LSM14B protein (*Figure 6I*). Exon six contains the only predicted nuclear localization signal (RPPRRR) in LSM14B (*Figure 6H*) lying between the LSM and FDF domains. This suggests that RBPMS mediated AS might prevent nuclear shuttling and function of LSM14B in mRNA turnover.

## RBPMS controls the SMC transcription factor myocardin

Myocardin (MYOCD) is a key transcription factor in SMCs and cardiac muscle (*Li et al., 2003*). Skipping of *Myocd* exon 2a in cardiac muscle and proliferative SMCs produces a canonical mRNA encoding full length MYOCD (*Figure 7A*) (*Creemers et al., 2006*; *van der Veer et al., 2013*). Inclusion of exon 2a in differentiated SMCs introduces an in frame stop codon, and the N-terminally truncated Myocd isoform produced using a downstream AUG codon lacks the MEF2 interacting domain and is more potent in activating SMC-specific promoters and SMC differentiation (*Creemers et al., 2006*; *Imamura et al., 2010*; *van der Veer et al., 2013*). Significant changes in *Myocd* exon 2a splicing were not detected by rMATS, but manual inspection of RNA-Seq data and RT-PCR confirmed that *Myocd* exon 2a is more included in differentiated than proliferative PAC1 SMCs (*Figure 7B*) and its inclusion decreases upon RBPMS knockdown in differentiated PAC1 cells (*Figure 7B*, *Figure 2—figure supplement 2B*). Effects of lentiviral RBPMS-A overexpression were less clear (PSI - Dox = 37.7 ± 5.1, PSI + Dox = 47.1 ± 11.9), possibly related to the very low *Myocd* expression in the transduced lines. A conserved 200 nt region downstream of exon 2a contains two clusters of CAC motifs (*Figure 7C*), suggesting that RBPMS directly activates exon 2a splicing. To better understand the regulation of the *Myocd* exon 2a, we created a minigene of exon 2a and its flanking intronic regions. In transfected PAC1 cells exon 2a in the minigene was included to a basal level of ~30% and RBPMS-A or RBPMS-B significantly increased exon 2a inclusion (*Figure 7D*). The *Myocd* minigene was also tested in HEK293 cells. As expected, exon 2a was fully skipped in the non-smooth muscle cell line, but was highly responsive to RBPMSA and B with inclusion increasing to ~75% and 37%, respectively (*Figure 7E*). To test the role of the downstream CAC clusters we mutated each cluster (CAC to CCC mutations). Mutation of cluster 1 (mCAC) had a modest effect on activation by RBPMSA, although RBPMSB activity was severely impaired. Mutation of cluster two impaired activity of both RBPMS isoforms, while the combined mutations abolished all activation by RBPMS (*Figure 7E*). Insertion of a previously defined RBPMS site (Ube2v1 from *Farazi et al., 2014*) into the double mutant minigene restored activation by RBPMS-A, although RBPMS-B had minimal activity in this context (*Figure 7G*). To confirm binding of RBPMS to the CAC motifs, EMSAs and UV crosslinking were carried out with in vitro transcribed RNAs containing the wild-type and mutant CAC

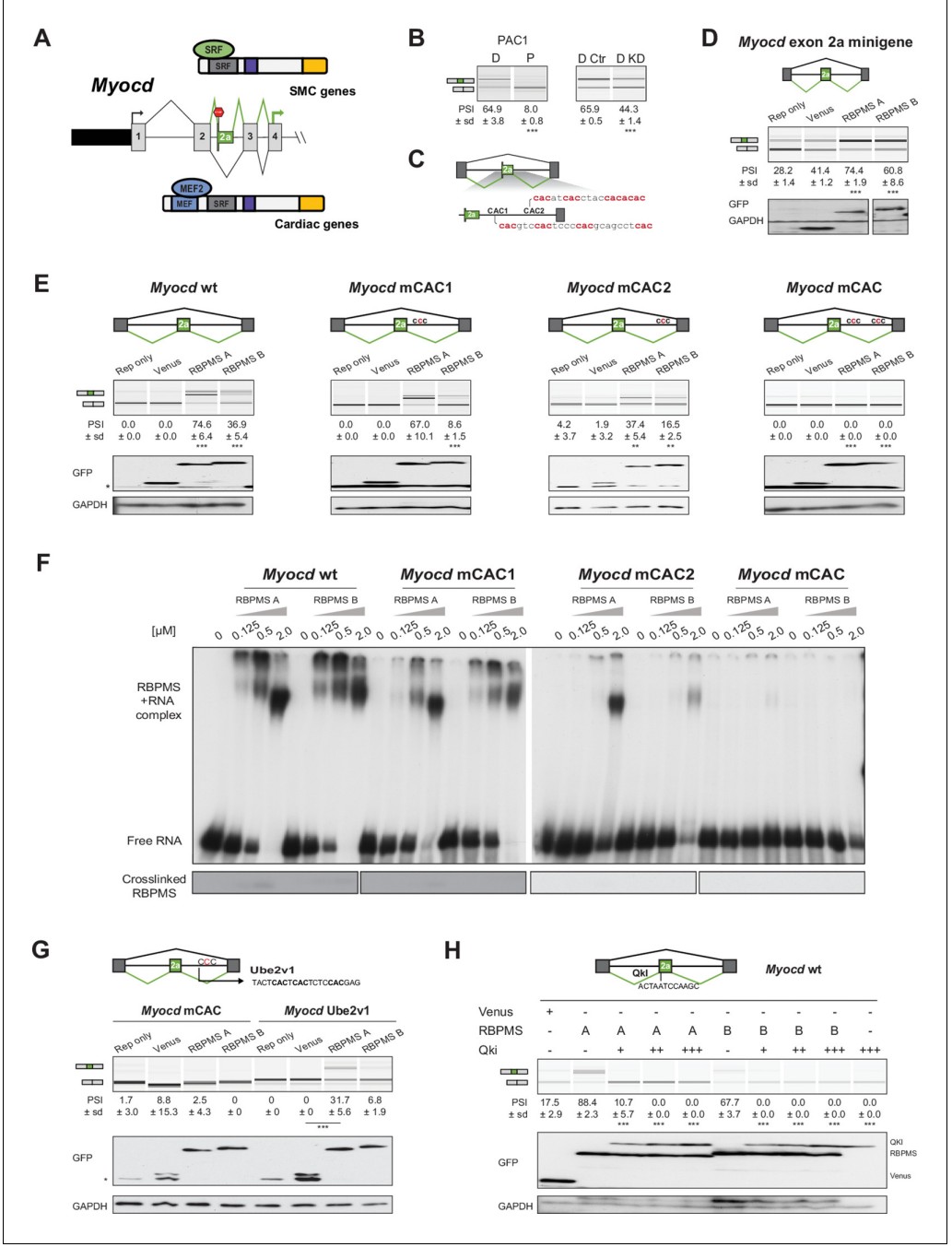

**Figure 7.** RBPMS controls splicing of the SMC transcription factor *Myocd* exon 2a. (**A**) Schematic of *Myocd* exon 2a ASE and its functional consequences upon MYOCD protein domains. (**B**) RT-PCR of *Myocd* exon 2a splicing in the PAC1 Differentiated (D) and Proliferative (P) PAC1 cells, left) and in differentiated PAC1 cells treated with control (D Ctr) or RBPMS siRNAs (D KD), right. (**C**) Schematic of the *Myocd* exon 2a splicing reporter. The two main clusters of CAC motifs, found downstream of exon 2a, are highlighted in red. (**D**) RT-PCRs of the effects of RBPMS A and B overexpression on *Myocd* exon 2a in PAC1 cells. Reporter only and Venus controls were tested alongside. Protein expression levels are shown in the western blot probing for GFP and GAPDH as a loading control. (**E**) RT-PCR of the effects of RBPMS A and B overexpression on Myocd exon 2a splicing reporter in HEK293. RBPMS CAC sites were mutated and response to RBPMS validated by RT-PCR. Schematic of the different mutant Myocd exon 2a splicing reporters are found at the top. RBPMS overexpression was confirmed by western blot using GAPDH as a loading control. (**F**) In vitro binding of RBPMS A and B to Myocd exon 2a downstream intron sequence in EMSA, top, and in UV-crosslink, bottom. In vitro transcribed wild-type and different mCAC RNAs were incubated with recombinant RBPMS (0.125 to 2 μM). (**G**) The UBE2V1 sequence identified to bind to

*Figure 7 continued on next page*

*Figure 7 continued*

RBPMS by PAR-CLIP (*Farazi et al., 2014*) was inserted into the mCAC Myocd reporter. Schematic of the mCAC Myocd exon 2a splicing reporter and the UBE2V1 sequence. RT-PCR of the effects of RBPMS A and B overexpression on Myocd reporter in HEK293 cells. Western blot probing for GFP and GAPDH as a loading control. (H) Effects of the co-expression of RBPMS and QKI, a Myocd exon 2a repressor, on Myocd exon 2a reporter in HEK293. Myocd 2a reporter schematic with QKI and potential RBPMS-binding sites indicated. RBPMS and QKI overexpression was confirmed by western blot against GFP and GAPDH as a loading control. In GFP western blots, '*' indicates the GFP product from the splicing reporter.

clusters (*Figure 7F*). RBPMS A and B were both able to bind to the wild-type RNA at similar affinities as indicated by EMSA (apparent Kd ~0.5 µM) (*Figure 7F* upper panels). Binding was modestly reduced by mutation of the first cluster (Kd ~0.5–2.0 µM), more severely affected by mutation of the second cluster (Kd ~2.0 µM), and eliminated by combined mutation of both clusters. UV crosslinking of RBPMS was inefficient with only a very faint signal evident with wild type, but not mutant, probes (*Figure 7F* lower panels). These data therefore indicate that RBPMS directly activates Myocd exon 2a inclusion via downstream CAC clusters.

Myocd exon 2a is repressed by binding of the RBP QKI to the 5' end of the exon (*van der Veer et al., 2013*). QKI is expressed more highly in proliferative SMCs (*Llorian et al., 2016*; *van der Veer et al., 2013*) suggesting that RBPMS and QKI could act antagonistically on AS events during phenotypic switching. To test for functional antagonism, we co-transfected the Myocd minigene with the two RBPs in HEK293 cells (*Figure 7H*). QKI strongly antagonized RBPMS activation of exon 2a inclusion, even at low concentrations, showing it to be the dominant regulator (*Figure 7H*). Thus, splicing of Myocd exon 2a, and thereby the transcriptional activity of Myocd, is under the antagonistic control of RBPs that are preferentially expressed in differentiated (RBPMS) or proliferative (QKI) SMCs.

## Discussion

### Using super-enhancers to identify master splicing regulators

By focusing on RBPs whose expression is driven by super-enhancers (*Jangi and Sharp, 2014*) we identified RBPMS as a key regulator of the differentiated SMC AS program, with many of the criteria expected of a master regulator: i) it is highly up-regulated in differentiated SMCs (*Figure 1*, *Figure 1—figure supplements 1–3*). Indeed, single cell RNA-Seq identified *Rbpms* as part of a transcriptome signature of contractile mouse aorta SMCs cells (*Dobnikar et al., 2018*); ii) changes in RBPMS activity appear to be solely responsible for 20% of the AS changes between differentiated and proliferative PAC1 cells (*Figure 3*); iii) RBPMS target splicing events are enriched in functionally coherent groups of genes affecting cell-substrate adhesion and the actin cytoskeleton, which are important for SMC cell phenotype-specific function (*Figure 5*); iv) it regulates splicing and activity of other post-transcriptional regulators in SMCs (*Figure 6*), and v) it regulates splicing of the key SMC transcription factor MYOCD (*Figure 7*) to an isoform that promotes the contractile phenotype (*van der Veer et al., 2013*).

RBPMS is reported to be a transcriptional co-activator (*Fu et al., 2015*; *Sun et al., 2006*). However, we did not observe changes in expression levels of SMC marker genes upon RBPMS knockdown or overexpression and changes in RNA abundance were outnumbered by regulated AS events (*Figure 2*). As a splicing regulator RBPMS can only affect actively transcribed genes, so it is unlikely to be sufficient to initially drive SMC differentiation. Notably, we identified RBPMS using super-enhancers mapped in adult human tissues (*Hnisz et al., 2013*). Combined with the ability of RBPMS to promote AS patterns characteristic of fully differentiated tissue SMCs, such as in *Cald1*, *Fermt2* and *Tns1* (*Figure 3F,G*), this suggests that RBPMS plays a key role in maintaining a mature adult SMC phenotype. Consistent with a role in promoting a fully differentiated state, in other cell types RBPMS has anti-proliferative tumor-suppressive activity (*Fu et al., 2015*; *Hou et al., 2018*; *Rastgoo et al., 2018*).

## RBPMS is an alternative splicing regulator

RBPMS and RBPMS2 (referred to as Hermes in *Xenopus*) are present across vertebrates, while the related proteins *Drosophila* Couch Potato and *C. elegans* MEC-8 bind to similar RNA sequences (*Soufari and Mackereth, 2017*). RBPMS and RBPMS2 can localize to the cytoplasm and nucleus, but apart from transcriptional co-regulation (*Fu et al., 2015*; *Sun et al., 2006*) most attention has been paid to cytoplasmic roles in mRNA stability (*Rambout et al., 2016*), transport (*Hörnberg et al., 2013*) and localization in cytoplasmic granules (*Farazi et al., 2014*; *Furukawa et al., 2015*; *Hörnberg et al., 2013*). RBPMS2 interacts with eukaryote elongation factor-2 (eEF2) in gastrointestinal SMCs, suggesting translational control (*Sagnol et al., 2014*). MEC-8 was reported to regulate splicing of Unc-52 in *C. elegans* (*Lundquist et al., 1996*), but otherwise RBPMS family members have not been reported to regulate splicing. PAR-CLIP with over-expressed RBPMS in HEK293 cells revealed its preferred binding site, but accompanying mRNA-Seq did not identify regulated AS events associated with PAR-CLIP peaks (*Farazi et al., 2014*). Using rMATS to re-analyze the RNA-Seq data of Farazi et al, we found a small number of regulated AS events, including the *Flnb* H1 exon (*Figure 4C–F*). We readily detected specific binding of RBPMS to target RNAs by EMSA, but UV crosslinking was inefficient, even with purified protein and in vitro transcribed RNA (*Figures 4F,J* and *7F*). This suggests that CLIP might significantly under-sample authentic RBPMS-binding sites. Nevertheless, the functional binding of RBPMS to *Flnb*, *Tpm1* and *Myocd* RNAs (*Figures 4* and *7*), the strong enrichment of dual CAC motifs with RBPMS-regulated exons (*Figure 4A*), the associated requirement for RBPMS RNA binding and dimerization (*Figure 4—figure supplement 3*), and distinct positional signatures for activation and repression of splicing (*Figure 4A,B*), indicate that RBPMS acts widely to directly regulate splicing in differentiated SMCs. While earlier reports have shown that RBPMS can act in other steps in gene expression, our data provide the strongest evidence to date for a widespread molecular function of RBPMS as a splicing regulator.

Cell-specific splicing programs are usually driven by more than one regulatory RBPs. A number of splicing regulatory RBPs are known to promote the proliferative SMC phenotype, including QKI, PTBP1, and SRSF1 (*Llorian et al., 2016*; *van der Veer et al., 2013*; *Xie et al., 2017*). The extent to which these proteins coordinately regulate the same target ASEs remains to be established. RBPMS and QKI antagonistically regulate at least two targets in addition to *Myocd* (*Figure 7*). The *Flnb* H1 exon is activated by RBPMS (*Figure 2—figure supplement 3A*, *Figure 4*) and repressed by QKI (*Li et al., 2018*), while the penultimate exon of *Smtn* is repressed by RBPMS (*Figure 2—figure supplement 3A*) but activated by QKI (*Llorian et al., 2016*). Moreover, QKI binding motifs are significantly associated with exons regulated during SMC dedifferentiation, and with exons directly regulated by RBPMS overexpression or knockdown (data not shown). It is therefore an interesting possibility that RBPMS and QKI might target a common set of ASEs perhaps acting as antagonistic master regulators of differentiated and proliferative SMC phenotypes. The two RBPs show reciprocal regulation of expression levels during SMC dedifferentiation (*Figure 1—figure supplement 2B,C*), which combined with antagonistic activities could lead to switch like changes in many ASEs. For *Myocd* splicing, QKI appears to have dominant activity, driving skipping of exon 2a even when RBPMS is present at higher levels (*Figure 7G*), so full inclusion of exon 2a requires the presence of RBPMS and absence of QKI. The logic of this regulatory input can explain inclusion of *Myocd* exon 2a in SMC and skipping in cardiac muscle and is consistent with the observation that RBPMS is super-enhancer associated both in vascular SMCs and heart left ventricle, while QKI is super-enhancer associated in left ventricle but not differentiated SMCs. Notably, QKI also controls a significant fraction of ASEs regulated during myogenic differentiation of skeletal muscle cells, but it promotes differentiated myotube AS patterns (*Hall et al., 2013*), in contrast to its promotion of dedifferentiated splicing patterns in SMCs.

## Activity of RBPMS isoforms

*RBPMS* was originally characterized as a human gene that encoded multiple isoforms of an RBP (*Shimamoto et al., 1996*), but differential activity of common RBPMS isoforms has not previously been reported. Human RBPMSA and B have similar activity for co-regulation of AP1 transcriptional activity (*Fu et al., 2015*) although a third human-specific isoform had lower activity. We found differential activity of RBPMSA and B upon some ASEs (*Figures 4* and *7*). In general, RBPMSA has higher activity particularly for repressed targets (*Figure 4F*, *Figure 4—figure supplements 1A* and

*2A*). The differential activity was seen with similar levels of overall expression, although we could not rule out the possibility of variation in nuclear levels as GFP-tagged RBPMS was predominantly cytoplasmic. Nevertheless, some ASEs were differentially responsive to RBPMSA or B while others responded equally (e.g. compare *MPRIP* with *ACTN1*, *Figure 4—figure supplement 1A*). The differential activity upon *Tpm1* splicing could not be accounted for by differences in RNA binding by RBPMSA or B (*Figure 4F,H*). Therefore, it is possible that RBPMS repressive function depends on other interactions mediated by the 20 amino acid RBPMSA C-terminal (*Figure 1C*). While the RRM domain is sufficient for RNA binding and dimerization in vitro (*Sagnol et al., 2014*; *Teplova et al., 2016*), previous studies have shown the extended C terminal region downstream of the RRM domain to be involved in several aspects of RBPMS/RBPMS2 function including granular localization in retinal ganglion cells (*Hörnberg et al., 2013*) and interaction with cFos in HEK293 cells (*Fu et al., 2015*). In vitro assays suggested that the C-terminal region increases RNA-binding affinity and possibly the oligomeric state of RBPMSA (*Farazi et al., 2014*) and the C-terminal 34 amino acids of *Xenopus* RBPMS2 is required for binding to *Nanos1* RNA in vivo (*Aguero et al., 2016*). In preliminary studies we have also found the C-terminal region to be essential for splicing regulation (data not shown). Future work will aim to address the mechanisms of RBPMS splicing activation and repression as well as isoform-specific differential activity.

## RBPMS targets numerous mRNAs important for SMC function

SMC phenotypic switching involves interconversion between a contractile phenotype and a motile, secretory, proliferative state (*Owens et al., 2004*). The actin cytoskeleton and its connections to the extracellular matrix (ECM) via focal adhesions are central to the function of both cell states, but with contrasting outcomes: tissue-wide contraction or independent movement of individual cells. RBPMS-mediated AS plays a major role in remodelling the actin cytoskeleton, the integrin adhesome (*Horton et al., 2015*) and ECM components in the two cell states. This is reflected in the similar GO terms shared by RBPMS-regulated AS events and the entire PAC1 cell AS program and in the morphological changes, including actin fiber reorganization, observed upon RBPMS knockdown in PAC1 cells (*Figure 5*, *Figure 5—figure supplements 1* and *2*). The importance of many of the RBPMS target genes to SMC function is further indicated by their proximity to super-enhancers in SMC tissues (*Figure 5B,C*). Indeed, three targets - ACTN1, FLNB and TNS1 - are all super-enhancer associated, interact directly with both actin and integrins and are components of distinct axes of the consensus integin adhesome (*Horton et al., 2015*). ACTN1 is a major hub in the network of proteins affected by RBPMS (*Figure 5C*). The AS event in ACTN1 produces a functional $Ca^{2+}$-binding domain in the motile isoform, but lack of $Ca^{2+}$ binding in the differentiated isoform stabilizes ACTN1 containing structures in contractile cells (*Waites et al., 1992*). Many other RBPMS-regulated AS events have not previously been characterized. Focal adhesion complexes and the integrin adhesome are mechanosensitive complexes that connect the cytoskeleton and ECM, and are hubs of regulatory tyrosine phosphorylation signalling. We found a small network of RBPMS-regulated AS events in the receptor tyrosine phosphatase PTPRF (*Figure 2E*) and two interacting proteins PPFIA1 and PPFIBP1 (*Figure 2—figure supplement 3A*). Another focal adhesion associated target is PIEZO1 (*Figure 2E*), a mechanosensitive ion channel that is important in SMCs during arterial remodelling in hypertension (*Retailleau et al., 2015*). An RBPMS repressed exon lies within the conserved Piezo domain immediately adjacent to the mechanosensing 'beam' (*Liang and Howard, 2018*). ECM components affected by RBPMS include fibronectin (FN1), which interacts directly with integrins, and HSPG2 (*Figure 2—figure supplement 3A*). Notably, HSPG2 (also known as Perlecan) is the basement membrane heparan sulfate proteoglycan that is the identified splicing target of MEC-8 in *C. elegans* (*Lundquist et al., 1996*).

In addition to direct regulation of numerous functionally related targets, by directly targeting transcriptional and post-transcriptional regulators RBPMS has the potential for more widespread action (*Figures 6* and *7*). RBPMS-regulated events in MBNL1 and 2 modulate their splicing regulatory activity (*Sznajder et al., 2016*; *Tabaglio et al., 2018*; *Tran et al., 2011*). Changes in secondary AS targets are challenging to observe in a short duration overexpression experiment. Nevertheless, the modest change in *NCOR2* splicing appears to be attributable to an RBPMS-induced switch to less active MBNL isoforms (*Figure 6F*). The regulated event in LSM14B (*Figure 6G–I*) also has the potential to regulate mRNA stability, or an as yet uncharacterized nuclear role of LSM14B (*Kırlı et al., 2015*).

We also identified the SMC transcription factor as a direct target of RBPMS (*Figure 7*). A similar role has been shown for the proposed myogenic AS master regulator RBM24 (*Jangi and Sharp, 2014*), which stabilizes mRNA of the transcription factor Myogenin by binding to its 3'UTR (*Jin et al., 2010*). Similarly, by activating inclusion of *Myocd* exon 2a (*Figure 7*), RBPMS directs production of a Myocardin isoform that more potently promotes the differentiated SMC phenotype (*van der Veer et al., 2013*). Additional effects upon transcription could also be conferred by RBPMS activation of the FLNB H1 exon (*Figure 2—figure supplement 3A*, *Figure 4*). FLNB is primarily an actin binding and adhesion protein, but inclusion of the H1 hinge domain allows nuclear localization and antagonism of the transcription factor FOXC1 in epithelial cells (*Li et al., 2018*). FOXC1 and FOXC2 are expressed at higher levels in adult arteries than any other human tissue (*GTEx Consortium, 2013*). The modulation by RBPMS of FLNB isoforms therefore provides another route for indirect transcriptome regulation. The importance of Filamin RNA processing in SMCs by adenosine to inosine editing of FLNA was also recently highlighted by the cardiovascular phenotypes arising from disruption of this editing (*Jain et al., 2018*). A number of splicing regulators also influence miRNA processing (*Michlewski and Cáceres, 2019*), so it is an interesting possibility that RBPMS might affect processing of SMC miRNAs such as miR143-145 (*Boettger et al., 2009*; *Cordes et al., 2009*).

RBPMS2 plays an important role in SMCs of the digestive tract. The *RBPMS2* gene is associated with super-enhancers in stomach smooth muscle (*Supplementary file 1*) and is expressed early in visceral SMC development and at lower levels in mature cells (*Notarnicola et al., 2012*). Ectopic RBPMS2 overexpression led to loss of differentiated contractile function *via* translational upregulation of *Noggin* (*Notarnicola et al., 2012*; *Sagnol et al., 2014*). This contrasts with our observations that RBPMS exclusively promotes differentiated SMC AS patterns. RBPMS2 is expressed at low levels in PAC1 and primary aorta SMCs (*Figure 1*) and its knockdown was without effect. Nevertheless, ectopic expression of RBPMS2 in PAC1 or HEK293 cells promoted differentiated SMC AS patterns in a similar manner to RBPMSA (*Figure 4*, *Figure 4—figure supplements 1* and *2*), suggesting that RBPMSA and RBPMS2 have intrinsically similar molecular activities. The reasons for the apparent discrepancy between the promotion by RBPMS2 of differentiated SMC splicing patterns, but de-differentiated visceral SMC phenotypes, remain to be resolved. Possible explanations include variations in cell-specific signalling pathways, pre-mRNA and mRNA targets, interacting protein partners, post-translational modifications and subcellular localization, all of which could differentially modulate RBPMS and RBPMS2 activity in different SMC types.

In conclusion, our data vindicate the proposal that tissue-specific AS master regulators might be identified by the association of their genes with superenhancers (*Jangi and Sharp, 2014*), paving the way for the identification of further such regulators in other tissues. While our data suggest that RBPMS has a critical role in SMCs, it is likely to play important roles in other cell types where its expression is also super-enhancer driven, including cardiac muscle and embryonic stem cells. Our approach aimed to identify AS master regulators common to diverse smooth muscle types (vascular, bladder, stomach). However, SMCs show a great deal of diversity (*Fisher, 2010*), even within single blood vessels (*Cheung et al., 2012*). The splicing regulator Tra2 β is responsible for some splicing differences between tonic and phasic SMCs (*Shukla and Fisher, 2008*), and it is possible that other RBPs might act as master regulators of some of these specialized SMC types. Our future studies aim to understand the mechanisms of splicing regulation by RBPMS, the role of the RBPMS-regulated splicing program in controlling different aspects of SMC phenotype and the potential role of subversion of this program in cardiovascular diseases. While the changes in actin organization observed upon RBPMS knockdown were striking (*Figure 5*), it will be important to directly test various functional readouts of SMC physiology related to actin function and cell adhesion, including cell motility and agonist-induced cell contraction. VSMCs can be generated by controlled differentiation from human embryonic stem cells (*Cheung et al., 2012*). In combination with the use of genomic safe harbors to allow inducible protein or shRNA expression, these cells provide an ideal model system in which to test the effects of RBPMS on VSMC differentiation and function. In the longer term, conditional knockout mouse models will provide invaluable insights into the role of RBPMS in vivo.

# Materials and methods

## Key resources table

| Reagent type (species) or resource | Designation | Source or reference | Identifiers | Additional information |
|---|---|---|---|---|
| Gene (*R. norvegicus*) | RBPMSA | | NCBI: XM_006253240.2; XP_006253302.1 | See Material and methods |
| Gene (*R. norvegicus*) | RBPMSB | | NCBI: NM_001271244.1; NP_001258173.1 | See Material and methods |
| Gene (*R. norvegicus*) | RBPMS2 | | NCBI: NM_001173426.1; NP_001166897.1 | See Material and methods |
| Cell line (*R. norvegicus*) | Rat PAC1 pulmonary artery SMCs | *Rothman et al., 1992* PMID: 1333373 | PAC1 | Cells from the original derivation of the line |
| Cell line (*H. sapiens*) | HEK293 | Lab stock | HEK293 | |
| Cell line (*H. sapiens*) | HEK293T | Lab stock | HEK293T | |
| Transfected construct (*R. norvegicus*) | effectors and minigenes | this study | | See Material and methods and *Supplementary file 6* |
| Biological sample (*R. norvegicus*) | Aorta tissue | this study | T | See Material and methods |
| Biological sample (*R. norvegicus*) | Dispersed VSMC | this study | SC | See Material and methods |
| Biological sample (*R. norvegicus*) | Passage 0 | this study | P0 | See Material and methods |
| Biological sample (*R. norvegicus*) | Passage 9 | this study | P9 | See Material and methods |
| Antibody | Mouse monoclonal anti-ACTA2 | Agilent/Dako | Agilent/Dako: M0851 | See *Supplementary file 6* |
| Antibody | Rabbit monoclonal anti-CALD1 | Abcam | Abcam: 32330 | See *Supplementary file 6* |
| Antibody | Rabbit polyclonal ani-GAPDH | Santa Cruz | Santa Cruz: sc-25778 | See *Supplementary file 6* |
| Antibody | Rabbit polyclonal anti-LSM14b | Sigma Aldrich | Sigma Aldrich: HPA041274 | See *Supplementary file 6* |
| Antibody | Rabbit polyclonal anti-MBNL1 | in-house antibody | NA | See *Supplementary file 6* |
| Antibody | Rabbit polyclonal anti-MBNL2 | Santa Cruz | Santa Cruz: sc-134813 | See *Supplementary file 6* |
| Antibody | Rabbit polyclonal anti-RBPMS | Sigma Aldrich | Sigma Aldrich: HPA056999 | See *Supplementary file 6* |

*Continued on next page*

*Continued*

| Reagent type (species) or resource | Designation | Source or reference | Identifiers | Additional information |
|---|---|---|---|---|
| Antibody | Rat monoclonal anti-TUBULIN | Abcam | Abcam: ab6160 | See *Supplementary file 6* |
| Antibody | Mouse monoclonal anti-FLAG | Sigma Aldrich | Sigma Aldrich: F-1804 | See *Supplementary file 6* |
| Antibody | Rabbit polyclonal anti-GFP | Invitrogen | Invitrogen: A-11122 | See *Supplementary file 6* |
| Recombinant DNA reagent | pEGFP-C1 | Clontech | GenBank Accession: U55763 | in vivo transfection |
| Recombinant DNA reagent | pCI-neo-3x-FLAG | *Rideau et al., 2006* PMID: 16936729 | | in vivo transfection |
| Recombinant DNA reagent | pET21d | Novagen | Novagen: 69743-3 | protein expression in *E. coli* |
| Recombinant DNA reagent | pCAGGs-EGFP | this lab | *Wollerton et al., 2004* | minigene reporter; see Material and methods |
| Recombinant DNA reagent | pT2 | this lab | *Gooding et al., 2013* | minigene reporter; see Material and methods |
| Recombinant DNA reagent | ACTN1 reporter | this lab | *Gromak et al., 2003* | minigene reporter; see Material and methods |
| Recombinant DNA reagent | pDONR221 | Invitrogen | Invitrogen: 12536017 | GATEWAY cloning |
| Recombinant DNA reagent | pINDUCER 22 | *Meerbrey et al., 2011* PMID: 21307310 | pINDUCER 22 | lentivirus production; see Material and methods |
| Recombinant DNA reagent | pINDUCER 22-RBPMSA | This study | pInducer22 -RBPMSA | See Materials and methods |
| Recombinant DNA reagent | pGEM4Z | Promega | Promega: P2161 | in vitro transcription templates; see Material and methods |
| Sequence-based reagent | oligonucleotides | Sigma Aldrich | | See *Supplementary file 6* |
| Sequence-based reagent | Control siRNA | Dharmacon | C2 custom siRNA | See Material and methods |
| Sequence-based reagent | RBPMS siRNA | Stealth siRNAs Thermo Fisher Scientific | RSS363828; KD1 | See Material and methods |
| Sequence-based reagent | RBPMS siRNA | Stealth siRNAs Thermo Fisher Scientific | RSS363829; KD2 | See Material and methods |
| Sequence-based reagent | RBPMS siRNA | Stealth siRNAs Thermo Fisher Scientific | RSS363830; KD3 | See Material and methods |
| Sequence-based reagent | MBNL1 siRNA | Dharmacon | THH2 siRNA; *Gooding et al., 2013* | See Material and methods |

*Continued on next page*

*Continued*

| Reagent type (species) or resource | Designation | Source or reference | Identifiers | Additional information |
|---|---|---|---|---|
| Sequence-based reagent | MBNL2 siRNA | Dharmacon | *Gooding et al., 2013* | See Material and methods |
| Peptide, recombinant protein | T7-FLAG-RBPMSA-H6 | this paper | | See Material and methods and *Supplementary file 6* |
| Peptide, recombinant protein | T7-FLAG-RBPMSB-H6 | this paper | | See Material and methods and *Supplementary file 6* |
| Chemical compound, drug | Phalloidin | Invitrogen | Alexa Fluor 488 phalloidin; A12379 | See Material and methods |
| Software, algorithm | ImageJ | https://imagej.nih.gov/ij/download.html | 1.8.0_172 | See Material and methods |
| Software, algorithm | FibrilTool plugin | *Boudaoud et al., 2014* PMID: 24481272 | | See Material and methods |
| Software, algorithm | AxioVision | Zeiss | v4.8.2 | See Material and methods |
| Software, algorithm | Rstudio | http://www.rstudio.com/ | R version 3.3.3 | See Material and methods |
| Other | DAPI staining | Thermo Fisher Scientific | ProLong Diamond Antifade Mountant with DAPI; P36966 | See Material and methods |

## Identification of potential master AS regulators

Locations of human super-enhancers (genome build Hg19) were taken from the data sets UCS-D_Aorta, UCSD_Bladder, BI_Stomach_Smooth_Muscle and BI_Skeletal_Muscle in *Hnisz et al. (2013)*. Associated genes were obtained using the UCSD Table Browser (GREAT version 3.0.0) with the Association rule: 'Basal +extension: 5000 bp upstream, 1000 bp downstream, 1000000 bp max extension, curated regulatory domains included'. The list of super-enhancer associated genes from the dbSUPER database consists of genes assigned more stringently to super-enhancers within a 50 kb window or where experimental verification was available (*Khan and Zhang, 2016*). We used the 1542 human RBPs from *Gerstberger et al. (2014)* to identify potential master AS regulators within each set of super-enhancer proximal genes (*Supplementary file 1*).

## DNA constructs

Coding sequences of rat *Rbpms* isoforms were PCR amplified from differentiated PAC1 cell cDNA and cloned into XhoI/EcoRI sites of the pEGFP-C1 vector (Clontech) and into EcoRI/XhoI sites of the pCI-neo-3x-FLAG vector (*Rideau et al., 2006*) to generate N-terminal Venus and 3xFLAG tagged in vivo overexpression constructs. The two major *Rbpms* isoforms identified were RBPMS A (XM_006253240.2/XP_006253302.1) and RBPMS B (NM_001271244.1/NP_001258173.1). RNA binding (K100E) and dimerization (R38Q and R38A/E39A) mutants, previously described in *Farazi et al. (2014)*, were generated by site-directed mutagenesis of RBPMS A. QKI construct has been described in a previous study (*Llorian et al., 2016*).

Splicing reporters of *Tpm1* exon three and *Actn1* exon NM and SM were described in *Gooding et al. (2013)*; *Gromak et al. (2003)*. *Myocd* exon 2a and *Flnb* exon H1 splicing reporters were obtained by PCR amplification of the target exons and respective flanking intron regions from genomic PAC1 DNA, approximately 250 bp upstream and downstream for *Myocd* and 500 bp for *Flnb*. PCR products were subsequently cloned into XhoI/EcoRV and NotI/SphI sites of pCAGGs-EGFP vector, which contains a GFP expression cassette with an intron inserted into its second codon (*Wollerton et al., 2004*). Point mutations of the CAC motifs in the splicing reporters were generated by PCR using oligonucleotides that contained A to C mutations. Intronic regions containing CACs

from *Tpm1*, *Flnb* and *Myocd* were PCR amplified and cloned into HindIII/EcoRI sites of pGEM4Z (Promega) for in vitro transcription. DNA constructs were confirmed by sequencing. All the oligonucleotides used for cloning and mutagenesis are found in *Supplementary file 6*.

## Cell culture, transfection and inducible lentiviral cells

All cell lines were tested for mycoplasma contamination by RNA capture ELISA, and tested negative. Rat PAC1 pulmonary artery SMCs (*Rothman et al., 1992*) were grown to a more differentiated or proliferative state as described in *Llorian et al. (2016)*. HEK293 and HEK293T cells were cultured following standard procedures. *Rbpms* siRNA mediated knockdown in PAC1 cells was performed as in *Llorian et al. (2016)*. Briefly, $10^5$ differentiated PAC1 cells were seeded in a six well plate. After 24 hr, cells were transfected using oligofectamine reagent (Invitrogen) and 90 pmols of Stealth siRNAs from Thermo Fisher Scientific (siRNA1: RSS363828, GGCGGCAAAGCCGAGAAGGAGAACA). A second treatment was performed after 24 hr using lipofectamine2000 (Thermo Fisher) and siRNA at the same concentration of the first treatment. Total RNA and protein were harvested 48 hr after the second knockdown. C2 scrambled siRNA was used as a control in the knockdown experiments (Dharmacon, C2 custom siRNA, AAGGUCCGGCUCCCCCAAAUG). To assess morphological changes in PAC1 cells upon RBPMS knockdown, the last siRNA treatment with lipofectamine2000 was repeated 48 hr after the second knockdown. PAC1 cells were then assessed 48 hr after the last treatment (120 hr after first siRNA treatment).

For *Mbnl1* and *Mbnl2* siRNA knockdown, the *Mbnl1* THH2 siRNA (CACGGAAUGUAAAUUUGCA UU) and *Mbnl2* specific siRNA (Dharmacon, GAAGAGUAAUUGCCUGCUUUU) were used (*Gooding et al., 2013*).

3xFLAG N-terminally tagged rat RBPMSA cDNA was cloned into pInducer22 (*Meerbrey et al., 2011*) using the Gateway system. Generation of stable PAC1 cell lines with pInducer22 vector only (LV) or pInducer22-RBPMSA was done as follows. Lentiviral particles were produced in HEK293T cells by transient transfection using 30 μl of Mirus TransIT-lenti (MIR6604), 7 μg 3xFLAG tagged RBPMSA cDNA and 0.75 μg of the packaging plasmids gag, pol, tat and VSV-G transfecting $2 \times 10^6$ cells per 10 cm dish. After 24 hr the medium was transferred to 4°C and replaced with fresh medium. After a further 24 hr the medium was removed, pooled with the first batch, spun at 1000 g for 5 min and filtered through 0.45 micron PVDF filter. Lentiviral particles were diluted 1:2 with fresh DMEM medium containing Glutamax, 10% FBS and 16 μg/ml polybrene and used to replace the medium on PAC1 cells plated 24 hr earlier at $10^4$/35 mm well, setting up two wells for pInducer22 vector only and six wells for RBPMSA. Fresh medium was added 24 hr later and the populations amplified as necessary. To induce expression of 3xFLAG RBPMSA the cells were plated at $4 \times 10^5$ cells per 35 mm well ±1 μg/ml doxycycline harvesting RNA and protein 24 hr later.

For transient transfections of HEK293 cells with splicing reporters and effectors, lipofectamine 2000 reagent was used and cells harvested 48 hr after transfection. To verify knockdown and transfection efficiency, total cell lysates were obtained by directly adding protein loading buffer to the cells. Lysates were run on a SDS-PAGE, followed by western blot against RBPMS and loading controls. See *Supplementary file 6* for information on the antibodies used in this study. To monitor changes in splicing and mRNA abundance, RNA was extracted using TRI reagent (Sigma) according to manufacturer's instructions, DNase treated with Turbo DNA-free kit (Thermo Fisher) and cDNA synthesized, as described below, followed by PCR and QIAxcel or qRT-PCR analysis.

## qRT-PCR and RT-PCR

cDNA was prepared using 1 μg total RNA, oligo(dT) or gene-specific oligonucleotides and SuperScript II (Life technologies) or AMV RT (Promega) as described in manufacturer's protocol. qRT-PCR reactions were prepared with 50 ng of cDNA, oligonucleotides for detection of mRNA abundance and SYBER Green JumpStart Taq Ready Mix (Sigma). Three-step protocol runs were carried out in a Rotor-Gene Q instrument (QIAGEN). Analysis was performed in the Rotor-Gene Q Series Software 1.7 using the Comparative Quantitative analysis. To normalize the relative expression values, two housekeeper genes were included in each experiment (*Gapdh* and *Rpl32*) and their geometric mean used for normalization. Expression values were acquired from biological triplicates.

PCRs with 50 ng of the prepared cDNAs were carried out to detect the different mRNA splicing isoforms of the reporters using the oligonucleotides detailed in *Supplementary file 6*. For

visualization and quantification of the PSI values, PCR products were resolved in a Qiaxcel Advanced System (QIAGEN) and PSI calculated within the QIAxcel ScreenGel software. A minus RT cDNA, representative of each triplicate, and a no template PCR reactions were also included in all the experiments (data not shown). Statistical significance was tested as previously described for gene expression (paired Student t-test for lentiviral experiments). PSI values are shown as mean (%) ± standard deviation (sd). For lentiviral RBPMSA overexpression, ΔPSI values determined by RT-PCR were derived from at least three independent transductions, and no events responded to doxycycline treatment in cells transduced with the empty pINDUCER22 vector (*Figure 2—figure supplement 2A*). Statistical significance was tested by a two-tailed Student's t test, paired for lentiviral experiments and unpaired for all the others (*p<0.05, **p<0.01, ***p<0.001).

## Immunostaining

For immunodetection of RBPMS in PAC1 cells, differentiated and proliferative cells were grown on coverslips, fixed with 4% paraformaldehyde (PFA) for 5 min, rinsed with phosphate buffer saline (PBS) and permeabilized with 0.5% NP-40 for 2 min followed by PBS washes. Coverslips were incubated with blocking buffer (1% BSA in PBS) for 1 hr and incubated with RBPMS primary antibody diluted in blocking buffer for another hour. Coverslips were rinsed and secondary antibody in blocking buffer applied to the coverslips which were incubated for 1 hr. Coverslips were washed and mounted on ProLong Diamond Antifade with DAPI (Thermo Fisher Scientific). For staining of actin fibers, cell were incubated with Alexa Fluor 488 phalloidin (Invitrogen, A12379), which was added at the secondary antibody incubation step at a 1:1000 dilution. All the steps were carried out at room temperature. Images were acquired from a fluorescence microscope (Zeiss Ax10, 40X) attached to CCD AxioCam and analyzed on AxioVision (v4.8.2).

## Image analysis

All the imaging analyses were carried out using the ImageJ free software (https://imagej.nih.gov/ij/download.html). To calculate the actin anisotropy, the FibrilTool plugin in ImageJ was used (*Boudaoud et al., 2014*). Phalloidin stained images were divided into four regions of interest and analysis performed for each area. To calculate nucleus size, DAPI stained images were analyzed using the following ImageJ commands: Adjust color threshold/Make Binary/Analyze Particles. The area values (Pixels) and count of nucleus in the field were then obtained for each field. To calculate the average of the cell size, first the fraction of the field occupied by the ACTIN staining was calculated using ImageJ (Adjust color threshold/Make Binary/Measure). The percentage of the area of ACTIN staining was divided by the number of cells in the respective field to obtain the average of the cell size. Statistical significance was tested using Mann-Whitney-Wilcoxon Test (*p<0.05, **p<0.01, ***p<0.001). Statistical analyses and data visualization were carried out in RStudio (http://www.rstudio.com/).

## RNAseq analyses

Total RNA from three biological replicates of RBPMS knockdown in differentiated PAC1 and three populations of RBPMS inducible overexpression in proliferative PAC1 cells were extracted for RNA-seq. Cells were lysed with TRI-reagent and total RNA purified by Direct-zol purification column (Zymo Research) followed by DNase treatment. RNAseq libraries of polyA selected RNAs were prepared with NEBNext ultra II RNA library prep kit for Illumina. Both RNA and RNAseq libraries were checked for their qualities. Barcoded RNAseq libraries were then multiplexed across two lanes of an Illumina HiSeq4000 platform for sequencing on a 150 bp paired-end mode, providing around 60 million reads per sample.

To investigate AS changes in vascular smooth muscle dedifferentiation, rat aortas were isolated from 8 to 12 weeks old Wistar rats. Aortas were briefly treated for 30 min at 37°C with 3 mg/ml collagenase (Sigma C-0130) to help in cleaning away the adventitia. The tissue was finely chopped and either used directly to make tissue RNA or enzymatically dispersed to single cells. This was achieved by treating the tissue pieces with 5 ml 1 mg/ml elastase (Worthington Biochemical Corporation LS002292) for 30 min at 37°C and then 5 ml collagenase added for a further 1–2 hr. Cells were washed and counted and plated at $4 \times 10^5$ cells/ml in M199 media containing 10%FBS, 2 mM Glutamine and 100 U/ml Penicillin-Streptomycin in a suitable dish according to the cell number. To

promote the proliferative state, SMCs were 1:2 passaged switching to DMEM media containing Glutamax and 10% FBS and harvested at passage 9. For RNAseq, total RNA was harvested from three replicas, each a pool of 5 rats, from rat aorta tissue (T), enzymatically dispersed single cultured SMCs (SC), passage 0 (P0) and passage 9 (P9). Total RNA extraction was carried out with Tri-reagent (Sigma). Libraries for mRNAseq were prepared using Ribozero and TrueSeq kits and sequencing performed on a HiSeq2000 platform in a paired-end mode.

## mRNA abundance analysis

Read trimming and adapter removal were performed using Trimmomatic version 0.36 (*Bolger et al., 2014*). Reads were aligned using STAR version 2.5.2a (*Dobin et al., 2013*) to the Rat genome Rnor_6.0 release-89 obtained from Ensembl and RSEM package version 1.2.31 (*Li and Dewey, 2011*) was used to obtain gene level counts. mRNA abundance analysis, was carried out with DESeq2 package (version 1.18.1) (*Love et al., 2014*) within R version 3.4.1 (https://www.r-project.org/). Genes were considered to be differential expressed with p-adj less than 0.05 in the paired analysis (*Supplementary file 2*).

## AS analysis

rMATS v3.2.5 (*Shen et al., 2014*) was used for detection of differential alternative splicing. rMATS analysis was carried out allowing for new splicing event discovery using the flag novelSS 1. rMATS calculates the inclusion levels of the alternative spliced exons and classifies them into five categories of AS events (*Figure 2—figure supplement 1B*): skipped exons (SE), mutually exclusive exons (MXE), alternative 5' and 3' splice sites (A5SS and A3SS) and retained intron (RI). Before further analysis, the results from reads on target and junction counts were filtered to include only events with a total of read counts above 50 across the triplicates in at least one of the conditions compared. Removal of events with low counts discarded false positive events. Only ASE with an FDR less than 0.05 were considered significant and a minimal inclusion level difference of 10% imposed to significant AS events. Finally, to identify specific AS events, unique IDs were created using the AS type, gene name and the chromosomal coordinates of the regulated and flanking exons (*Supplementary file 3*).

For visualization of differentially spliced exons, sashimi plots were generated using rmats2sashimiplot (*Gohr and Irimia, 2019*). The sashimi plots show the RNAseq coverage reads mapping to the exon-exon junctions and PSI values from rMATS. Twenty eight ASE identified by rMATS in the RBPMS knockdown or overexpression were also validated by RT-PCR in the same manner as described in the qRT-PCR and RT-PCR section. ΔPSI predicted from the RNAseq analysis and the ΔPSI observed in the RT-PCR were then tested for a Pearson correlation in RStudio (http://www.rstudio.com/).

For comparison and visualization of the overlap between the genes with different mRNA abundance and the genes differentially spliced in the RBPMS knockdown, RBPMS overexpression and the PAC1 dedifferentiation, proportional Venn diagrams were made using BioVenn (*Hulsen et al., 2008*). Venn diagrams were also generated for the visualization of the common AS events across RBPMS knockdown, overexpression and the PAC1 or aorta tissue dedifferentiation datasets.

## RBPMS motif enrichment analysis

RBPMS motif enrichment analyses in the regulated cassette exons (SE) of RBPMS knockdown and overexpression, PAC1 and Aorta tissue dedifferentiation were performed using the toolkit MATT (*Gohr and Irimia, 2019*). Cassette exons in transcripts identified by rMATS with significant changes (FDR < 0.05 and |ΔPSI| > 10%) were used to test enrichment or depletion of $CACN_{1-12}CAC$ RBPMS recognition element (*Farazi et al., 2014*), against a background set of unregulated exons defined as events with FDR > 0.1 and |ΔPSI| < 5%. 250 bp of the flanking intronic regions were examined for RBPMS motif signals. The motif enrichment scores were first obtained using the test_regexp_enrich command of the Matt suite in the quant mode with statistical significance determined using a permutation test with 50,000 iterations. This module inherently divides the examined regions (exons or 250 bp of the introns) into thirds and provides positional information for the occurrence of the RBPMS motif.

RNA maps for distribution of the RBPMS motif were generated using the rna_maps command in the Matt suite. Here the unregulated set of exons was randomly downsampled to include a final background of 2000 events only. Additionally, the program was instructed to scan only 135 bp of the cassette exon and 250 bp at either end of the flanking introns. A sliding window of 31 was used for scanning the motif and the statistically significant regions (p<0.05) for enrichment or depletion were identified by the permutation tests specifying 1000 iterations.

## Gene ontology and PPI analysis

Enrichment for gene ontology terms in the differentially abundant and spliced genes were obtained from Gorilla (*Eden et al., 2009*). Two unranked lists of genes, target and background lists, were used for the GO analysis. The target list contained either the significant differential abundant genes (p-adj <0.05 and log2 Fold Change greater than one or less than −1) or the significant differentially spliced genes (FDR < 0.05 and ΔPSI threshold of 10%). Background gene lists were created for the PAC1 experiments and aorta tissue by selecting the genes whose expression were higher than 1 TPM in either of the conditions analyzed. For visualization, only the top five enriched GO terms of each category (biological process, cell component and molecular function) were shown in *Figure 5A* and *Figure 5—figure supplement 1*. The complete list of enriched terms in differentially spliced and abundant genes can be found in *Supplementary files 4* and *5*.

A Protein-protein interaction network for the genes differentially spliced by RBPMS was constructed using the STRING v10.5 database (*Szklarczyk et al., 2017*). RBPMS-regulated genes were obtained by merging two lists: i) overlap of genes concordantly differentially spliced in the RBPMS knockdown and PAC1 experiments and, ii) overlap of genes concordantly differentially spliced in the RBPMSA overexpression and the aorta tissue datasets that were shown to be regulated in the same range in both conditions. A cut off of ΔPSI greater than 10% was also applied to the RBPMS-regulated gene list, similar to the GO analysis. The human database was chosen for the analysis and the following parameters applied to the PPI network: confidence as the meaning of the network edges, experiments and database as the interaction sources and high confidence (0.700) as the minimum required interaction score. STRING functional enrichments, using the whole genome as statistical background, were also included for visualization. Human super-enhancer associated genes from *Supplementary file 1* were highlighted.

## Recombinant protein

Rat RBPMS A and B with a 3xFLAG N terminal tag were cloned into the BamHI/XhoI sites of the expression vector pET21d, for expression of recombinant RBPMS containing a T7 N-terminal tag and a His$_6$ C-terminal tag in *E. coli*. Recombinant RBPMS A protein was purified using Blue Sepharose six and HisTrap HP columns whereas RBPMS B was purified only through the latter, since low binding was observed to Blue Sepharose 6. The identity of purified recombinant proteins was confirmed by western blot (*Figure 4—figure supplement 4*) and mass mapping by mass spectrometry.

## In vitro transcription and binding

$\alpha^{32}$P-UTP labelled RNA probes were in vitro transcribed using SP6 RNA polymerase. For EMSAs, a titration of the recombinant RBPMS A and B (0, 0.125, 0.5 and 2 µM) was incubated with 10 fmol of in vitro transcribed RNA in binding buffer (10 mM Hepes pH 7.2, 3 mM MgCl$_2$, 5% glycerol, 1 mM DTT, 40 mM KCl) for 25 min at 30°C. After incubation, samples were run on a 4% polyacrylamide gel. For UV-crosslinking experiments, the same binding incubation was performed followed by UV-crosslink on ice in a Stratalinker with 1920 mJ. Binding reactions were then incubated with RNase A1 and T1 at 0.28 mg/ml and 0.8 U/ml respectively, for 10 min at 37°C. Prior to loading the samples into a 20% denaturing polyacrylamide gel, SDS buffer was added to the samples which were then heated for 5 min at 90°C.

## Statistical analysis

Analysis and quantification of RNAseq, RT-PCR and imaging experiments were described in their respective sections with further information of the tests used in the different experiments present in the figure legends. Graphics were generated in RStudio (http://www.rstudio.com/).

## Data availability

mRNAseq of RBPMS (knockdown and overexpression) and Aorta tissue dedifferentiation data from this study have been deposited in NCBI Gene Expression Omnibus (GEO) repository under GEO accession GSE127800, accession number GSE127799 and GSE127794, respectively.

## Acknowledgements

We thank Elisa Monzon-Casanova for helpful comments on the manuscript, Juan Mata, Sushma Grellscheid, Vasudev Kumanduri and Elisa Monzon-Casanova for help and advice on RNA-Seq data analysis, and Mark Carrington for the fluorescence microscope used in the imaging experiments in this study.

## Additional information

### Funding

| Funder | Grant reference number | Author |
| --- | --- | --- |
| British Heart Foundation | PG/16/28/32123 | Sanjay Sinha<br>Christopher WJ Smith |
| British Heart Foundation | FS/11/85/29129 | Adrian Buckroyd<br>Christopher WJ Smith |
| British Heart Foundation | FS/18/46/33663 | Sanjay Sinha |
| Wellcome | 092900/Z/10/Z | Christopher WJ Smith |
| Wellcome | 209368/Z/17/Z | Christopher WJ Smith |
| Conselho Nacional de Desenvolvimento Científico e Tecnológico | 206813/2014-7 | Erick E Nakagaki-Silva |

The funders had no role in study design, data collection and interpretation, or the decision to submit the work for publication.

### Author contributions

Erick E Nakagaki-Silva, Investigation, Writing—original draft; Clare Gooding, Miriam Llorian, Aishwarya G Jacob, Investigation, Writing—review and editing; Frederick Richards, Adrian Buckroyd, Investigation, Data interpretation; Sanjay Sinha, Funding acquisition, Writing—review and editing; Christopher WJ Smith, Conceptualization, Funding acquisition, Writing—original draft

### Author ORCIDs

Erick E Nakagaki-Silva (ID) https://orcid.org/0000-0002-6878-4409
Christopher WJ Smith (ID) https://orcid.org/0000-0002-2753-3398

### Decision letter and Author response

Decision letter https://doi.org/10.7554/eLife.46327.sa1
Author response https://doi.org/10.7554/eLife.46327.sa2

## Additional files

### Supplementary files

• Supplementary file 1. RBP genes associated with super-enhancers in human tissues.

• Supplementary file 2. Genes with significant changes in mRNA abundance in aorta dedifferentiation (T vs P9), PAC1 dedifferentiation (D Ctr vs P Ctr), RBPMS knockdown (D KD vs D Ctr) and RBPMS overexpression (P OE vs P Ctr).

- Supplementary file 3. Genes with significant changes in mRNA splicing in aorta dedifferentiation (T - P9), PAC1 dedifferentiation (D Ctr - P Ctr), RBPMS knockdown (D KD - D Ctr) and RBPMS overexpression (P OE - P Ctr).
- Supplementary file 4. GO terms significantly enriched in the genes differentially spliced in aorta dedifferentiation (T - P9), PAC1 dedifferentiation (D Ctr - P Ctr), RBPMS knockdown (D KD - D Ctr) and RBPMS overexpression (P OE - P Ctr).
- Supplementary file 5. GO terms significantly enriched in the genes with differential mRNA abundance.
- Supplementary file 6. Oligonucleotides and antibodies used in this study.
- Transparent reporting form

### Data availability

RNA-Seq data have been deposited as FASTQ files at Gene Expression Omnibus with the reference SuperSeries GSE127800. The separate experiments can be accessed as the SubSeries: (1) RNAseq analysis of primary differentiated rat aorta tissue compared to proliferative cultured cells (accession number: GSE127794) (2) RBPMS knockdown and overexpression in rat PAC1 pulmonary artery smooth muscle cells (SMCs) (accession number: GSE127799).

The following dataset was generated:

| Author(s) | Year | Dataset title | Dataset URL | Database and Identifier |
|---|---|---|---|---|
| Christopher WJ Smith | 2019 | RNA-seq analysis of rat smooth muscle cells | http://www.ncbi.nlm.nih.gov/geo/query/acc.cgi?acc=GSE127800 | NCBI Gene Expression Omnibus, GSE127800 |

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
