## [Decision Letter]

Thank you for submitting your article "Identification of RBPMS as a smooth muscle master splicing regulator via proximity of its gene with super-enhancers" for consideration by *eLife*. Your article has been reviewed by three peer reviewers, and the evaluation has been overseen by Douglas Black as the Reviewing Editor and James Manley as the Senior Editor. The following individual involved in review of your submission has agreed to reveal their identity: Juan Valcárcel (Reviewer #2).

The reviewers have discussed the reviews with one another and the Reviewing Editor has drafted this decision to help you prepare a revised submission.

Summary:

This study from Nakagaki-Silva and colleagues characterizes the RNA binding protein RBPMS as a key regulator of alternative splicing in vascular smooth muscle. The authors found that the *RBPMS* gene is associated with superenhancers active in smooth muscle cell (SMC) rich tissues, similar to other key smooth muscle effector genes. Using the PAC-1 cell line to model the transition between the differentiated (D) and proliferative (P) states of SMC's, they show that RBPMS is high in the D state but repressed upon transition to the P state, similar to several key markers of SMC differentiation and in parallel with changes in the alternative splicing of SMC transcripts. To assess the role of RBPMS in alternative splicing, they set up a series of RNAseq comparisons. Data sets were generated from control differentiated cells, D cells with RBPMS depleted by siRNA, and from proliferating cells transduced with RBPMS or control empty lentivirus. The lentiviral transgene was under tet control, so the two virally infected P cells were examined both plus and minus doxycycline to induce RBPMS. From these data, they identify large sets of splicing events that shift during the D to P transition, or in response to RBPMS depletion or over expression. Some of these splicing events are confirmed in RT/PCR assays. Splicing patterns changing upon differentiation have significant overlap with those responding to RBPMS overexpression, and especially RBPMS depletion. These comprise two of four clusters of splicing events that are altered between D and P, one cluster of RBPMS activated exons and one exhibiting apparent repression by the protein. These events and their clustering were confirmed in differentiating primary cells from Aorta.

The authors estimate that 20% of the splicing events specific to differentiated SMC are driven by RBPMS expression. RBPMS is known to recognize two closely spaced CAC motifs. Examining their exons that are repressed by RBPMS, these motifs were found to be enriched upstream and within the exon. For exons activated by the protein, CAC motifs are enriched downstream, in keeping with widely observed positional effects of splicing regulators. The authors confirm the direct regulation of particular exons using minigenes to show that the effects of RBPMS overexpression match the endogenous transcripts and that RBPMS mediated regulation is lost upon CAC motif mutation. The direct recognition of this motif is demonstrated in RNA mobility shift binding assays. GO and STRING analyses were used to examine the function of transcripts regulated by RBPMS. Although exons affected by RBPMS overexpression were not easily categorized, exons changing upon RBPMS knockdown showed enrichment for terms associated with cytoskeletal functions, cell adhesion, and actin-based processes. Many of these could be grouped into a protein-protein interaction network using STRING. Looking at particular transcripts with known function in SMC biology, the authors find RBPMS dependent splicing of the regulators Mbnl1 and Myocardin. Myocardin in particular is a transcription factor affecting both cardiac and smc development. The authors identify a RBPMS dependent exon in this gene whose inclusion creates the isoform specific to differentiated SMC. They show that RBPMS induces this exon through direct CAC motif binding, and that RBPMS action is blocked by Qk1, a previously identified regulator of Myocardin splicing.

Essential revisions:

The reviewers all found this to be a well-developed study that identifies a new regulator of smooth muscle development. The authors implicate RBPMS in SMC biology by a clever means and then go on to characterize it with well controlled experiments. Tissue-restricted splicing regulators are still relatively rare. All three reviewers felt that the study would be strengthened by additional data on the role of the RBPMS splicing program in muscle physiology. What does RBPMS serve to master regulate? Does it drive a detectable physiological change in cell determination or behavior? To what extent is RBPMS required to gain the smooth muscle phenotype?

It is not expected that the authors develop a mouse model to address this, but is there some cell culture assay that would shed additional light on the biology? For example, does gain/loss of RBPMS accelerate/retard the transition between the contractile and motile/proliferative smooth muscle cell phenotypes? The authors should assess how they might gain additional insights into the biological role of RBPMS.

---

## [Author Response]

Essential revisions:The reviewers all found this to be a well-developed study that identifies a new regulator of smooth muscle development. The authors implicate RBPMS in SMC biology by a clever means and then go on to characterize it with well controlled experiments. Tissue-restricted splicing regulators are still relatively rare. All three reviewers felt that the study would be strengthened by additional data on the role of the RBPMS splicing program in muscle physiology. What does RBPMS serve to master regulate? Does it drive a detectable physiological change in cell determination or behavior? To what extent is RBPMS required to gain the smooth muscle phenotype?It is not expected that the authors develop a mouse model to address this, but is there some cell culture assay that would shed additional light on the biology? For example, does gain/loss of RBPMS accelerate/retard the transition between the contractile and motile/proliferative smooth muscle cell phenotypes? The authors should assess how they might gain additional insights into the biological role of RBPMS.

We agree with the point raised by all referees of the need to characterize a phenotype associated with RBPMS. We were not previously able to discern an obvious phenotype under the conditions used for RNA analysis (typically 24-48 hr knockdown or overexpression), probably because this time-frame was insufficient to allow turnover of pre-existing protein isoforms and replacement by alternative isoforms. We now include new data (Figure 5D-F, Figure 5—figure supplements 2 and 3) showing that after sustained RBPMS knockdown for 120 hr in differentiated PAC1 cells there is a substantial reorganization of the actin cytoskeleton so that the cells now resemble proliferative PAC1 cells in overall cell morphology and actin cytoskeleton. This fits well with the enrichment among RBPMS targets of cytoskeletal and focal adhesion functions that we observe.

This is obviously only a start; in the future we aim to address the physiological roles of RBPMS using a number of model systems and different functional readouts. Ultimately, we plan to use conditional knockout mice but in the medium term we will use human stem cell derived vascular smooth muscle cells (VSMCs) using protocols developed by the Sinha lab. We can use genomic safe harbors to inducibly express either RBPMS or shRNAs targeting RBPMS, either prior to or following differentiation of VSMCs from neural crest or lateral plate mesoderm lineages. This is a better defined model cell system than the PAC1 cells, in which it should be possible to test for a range of cell functions affected by RBPMS, including cell motility and single cell contraction. However, this is beyond the scope of what we can achieve for revisions of the current manuscript. We have included a short section at the end of the Discussion addressing these future plans which, combined with the new cell phenotype data, we hope sufficiently addresses the requested Essential Revisions.